# The origin and evolution of open habitats in North America inferred by Bayesian deep learning models

Tobias Andermann [1,2,3] ✉, Caroline A. E. Strömberg[4], Alexandre Antonelli [2,3,5,6] & Daniele Silvestro [2,3,7,8] ✉

Some of the most extensive terrestrial biomes today consist of open vegetation, including temperate grasslands and tropical savannas. These biomes originated relatively recently in Earth's history, likely replacing forested habitats in the second half of the Cenozoic. However, the timing of their origination and expansion remains disputed. Here, we present a Bayesian deep learning model that utilizes information from fossil evidence, geologic models, and paleoclimatic proxies to reconstruct paleovegetation, placing the emergence of open habitats in North America at around 23 million years ago. By the time of the onset of the Quaternary glacial cycles, open habitats were covering more than 30% of North America and were expanding at peak rates, to eventually become the most prominent natural vegetation type today. Our entirely data-driven approach demonstrates how deep learning can harness unexplored signals from complex data sets to provide insights into the evolution of Earth's biomes in time and space.

The different types of vegetation and their spatial distribution form the biotic landscape on which other species, including all terrestrial animals, interact and evolve. From reconstructions of past vegetation (paleovegetation) at different sites, we know that vegetation continually evolves through time as a response to changes in climate[1,2], interactions with faunal communities[3], and large biological events such as mass extinctions[4].

Several major vegetation changes are documented in the fossil record, including the shift from ecosystems dominated by free-sporing plants to seed plants[5] and the radiation and ecological expansion of angiosperms[4,6,7], which today dominate the natural vegetation in most terrestrial biomes (excluding anthropogenic agricultural impacts). The most important vegetation change in the Cenozoic is arguably the origination and expansion of open grass-dominated habitats[8] at the expense of closed forest ecosystems[9]. Open grasslands were in recent history (i.e., prior to recent anthropogenic impacts) the most extensive

terrestrial biome on Earth, covering over 40% of the Earth's land surface[10]. The oldest confirmed presence of open-habitat grasses in North America dates back to the late Eocene (38–34 Ma), yet the fossil record indicates that these were rare elements of the vegetation and at the time did not form sizable open ecosystems[9,11]. Based on the currently available paleobotanical evidence, it is likely that open grass-dominated habitats first appeared as a novel biome relatively recently, sometime during the late Oligocene to early Miocene (~28–20 Ma[8,12]). Yet, the precise timings and dynamics of open grassland origination and expansion are still poorly understood and much debated (e.g., see Strömberg[9]).

Previous studies have produced paleovegetation reconstructions for individual sites based on the evaluation of (i) plant macrofossil assemblages (i.e., fossilized leaves, seeds, wood, or other plant organs); (ii) fossilized pollen; or (iii) phytoliths—microscopic silica bodies produced in plant cells with a high fossilization potential and

[1]Department of Organismal Biology, SciLifeLab, Uppsala University, Uppsala, Sweden. [2]Department of Biological and Environmental Sciences, University of Gothenburg, Gothenburg, Sweden. [3]Gothenburg Global Biodiversity Centre, Department of Biological and Environmental Sciences, University of Gothenburg, Gothenburg, Sweden. [4]Department of Biology & Burke Museum of Natural History and Culture, University of Washington, Seattle, WA, USA. [5]Department of Plant Sciences, University of Oxford, Oxford, UK. [6]Royal Botanic Gardens, Kew, Richmond, Surrey, UK. [7]Department of Biology, University of Fribourg, Fribourg, Switzerland. [8]Swiss Institute of Bioinformatics, Fribourg, Switzerland. ✉e-mail: tobias.andermann@ebc.uu.se; daniele.silvestro@unifr.ch

unique shapes, which can be attributed to specific vegetation components[13]. While such reconstructions can provide an accurate record of the vegetation at a given site, extrapolating these reconstructions to larger geographic scales and through time is hampered by the sparsity of fossil sites and the incompleteness of the fossil record, both in terms of taxonomic coverage as well as time series. Extrapolations are often done based on expert opinion, under consideration of paleoclimatic models and other sources of information[14–16], at the cost of reduced reproducibility and limited scalability.

Most modeling studies aimed at inferring past or future vegetation have been based on climate models and predefined tolerance limits for certain biome types[17,18]. However, additional types of data that are not normally integrated might considerably improve such modeling approaches, such as the associations between the faunal fossil record with the surrounding vegetation. For example, the relationship between grassland biomes and large grazing mammals, often identifiable by their hypsodont teeth, has long been used to infer the presence of grasslands[19,20] (but see Strömberg[21], and Dunn et al.[22]). Such information about plant-mammal interactions is commonly only used to infer the paleovegetation at individual fossil sites[9], although mammal fossil assemblages are sometimes used to validate and correct vegetation predictions made from climate-based models[18].

Mammal fossils are a useful data type because of their relatively rich record. Mammals are one of the paleontologically best studied groups with a relatively well-resolved fossil taxonomy, often allowing for precise identifications of fossil mammals down to genus or even species level. Many mammal fossil datasets are publicly available through large online databases (e.g., Carrasco et al.[23]). Yet, to our knowledge, no computational models exist that explicitly utilize this information to predict vegetation. This may be partly due to the fact that for most taxa, habitat associations are difficult to establish with confidence, particularly so for extinct taxa, and ambiguous for many mammal taxa that are not restricted to a single vegetation type.

To improve the modeling of past environments, we present here a Bayesian deep neural network (BNN) model that utilizes comprehensive information on mammal fossils, plant macrofossils, modeled paleoclimate and elevation estimates, as well as spatial and temporal coordinates, including tectonic movement (PALEOMAP). We train the model using independent vegetation datasets, including paleovegetation and modern vegetation information. To gather data on past vegetation, we compiled a dataset of 331 independent points with reconstructed paleovegetation, sampled across North America and throughout the last 30 million years (Myr, see Supplementary Data 1). In addition, we added information about past mammal-plant interactions by compiling a dataset of more than 5000 fossil occurrences for different mammal and plant taxa (Paleobiology Database, and Cenozoic Angiosperm database[24], see Supplementary Data 2), complemented by current occurrences of these taxa (Global Biodiversity Information Facility−GBIF). Importantly, our model does not require any prior assumptions on temperature tolerance limits or ecological interactions. Instead, it learns how these biotic and abiotic features can be mapped to a vegetation type within a supervised learning framework. This property provides great flexibility, as any available biotic or abiotic predictor can be added to the model; it is the model itself that decides which predictors are useful for the vegetation inference, based on the data provided. Once trained, the model can be applied to estimate the most likely vegetation for any given point in time and space (within the temporal and spatial scope covered by the training data), as well the uncertainty associated with the prediction. We implement our BNN model to infer past vegetation changes in North America throughout the last 30 Myr, focusing on the expansion of open vegetation.

## Results

### Model description

We implement a BNN model to predict vegetation through space and time (Figs. 1 and 2, see "Methods" for a more detailed description). We focus on two vegetation types "open vegetation" (including open grasslands, savannas and steppes, desert vegetation, and tundra) and "closed vegetation" (forests). Additional and more nuanced vegetation categories could be implemented for other systems if sufficient data were available for training. As features for this classification task we use biotic data, consisting of fossil occurrences of 100 selected mammal and plant taxa (see "Methods"), supplemented by current occurrences of these taxa. Further, we use several abiotic features including proxies for elevation, mean annual temperature, and precipitation through space and time[25], paleocoordinates[26], mean global temperature from oxygen isotope data[27], and mean global atmospheric $CO_2$ concentration estimates based on carbon isotope data from fossil soils and stomatal pore density of fossilized leaves[28].

Our deep neural network consists of two initial hidden layers, where distances derived from the raw mammal and plant fossil occurrence data (Fig. 1) are transformed into taxon-specific features (Fig. 2a, see "Feature generation" in "Methods" section for a more detailed explanation). These features, in combination with additional abiotic features, comprise the input data for the fully connected layers of a BNN classifier, which quantifies the probability of each vegetation type for a given point in space and time (Fig. 2b). Through this setup the model is trained to infer the vegetation type based on the measured distances to nearby taxon occurrences, in combination with the additional climatic and geographic features.

### Predicting past vegetation

To train our BNN vegetation classifier, we compiled a total of 331 paleovegetation reconstructions based on phytolith and pollen assemblages, paleosol data, and macrofossils from the peer-reviewed literature (see "Methods"), ranging in age from the beginning of the Oligocene (33.9 Ma) to the present (Supplementary Figs. 1 and 2). To further increase training data, we supplemented the paleovegetation data with data on current vegetation. Since current vegetation patterns are heavily influenced by human activity, we retrieved the SYNMAP Global Potential Vegetation data, representing the potential vegetation without human land alterations[29]. To find the model configuration that produced the best paleovegetation prediction accuracy, we tested a range of different model architectures, as well as different combinations of input data (Table 1). We applied a five-fold cross-validation approach to the training data when training each model; in this approach each model is trained five times on a different 80% of the input data, while using the remaining 20% as a test set. This allows to determine the overall prediction accuracy of the model by averaging the number of correctly predicted test set labels across all 5 test sets, comprising all available data. We calculated the prediction accuracy separately for paleovegetation and current vegetation, as well as a combined weighted mean of the two (Table 1, Supplementary Fig. 3, see "Methods" for more information).

The best model (model 1, Table 1) reached a prediction accuracy of 88.7% (88.8% paleo, 87.6% current). The model included biotic (taxon distances) and abiotic features and its architecture consisted of one layer containing 8 nodes, and no feature pooling (see "Methods" for more details). The prediction accuracy can be further improved by applying a posterior probability (PP) threshold to the class predictions, only making vegetation inferences for predictions that exceed this threshold. In separate sensitivity analyses, we show that applying PP thresholds, and thereby masking low-confidence predictions, leads to more accurate predictions of vegetation patterns (see Supplementary Discussion and Supplementary Figs. 4 and 5). The higher the PP threshold is set, the higher test accuracies can be reached, at the cost of an increasing number of test instances being predicted as

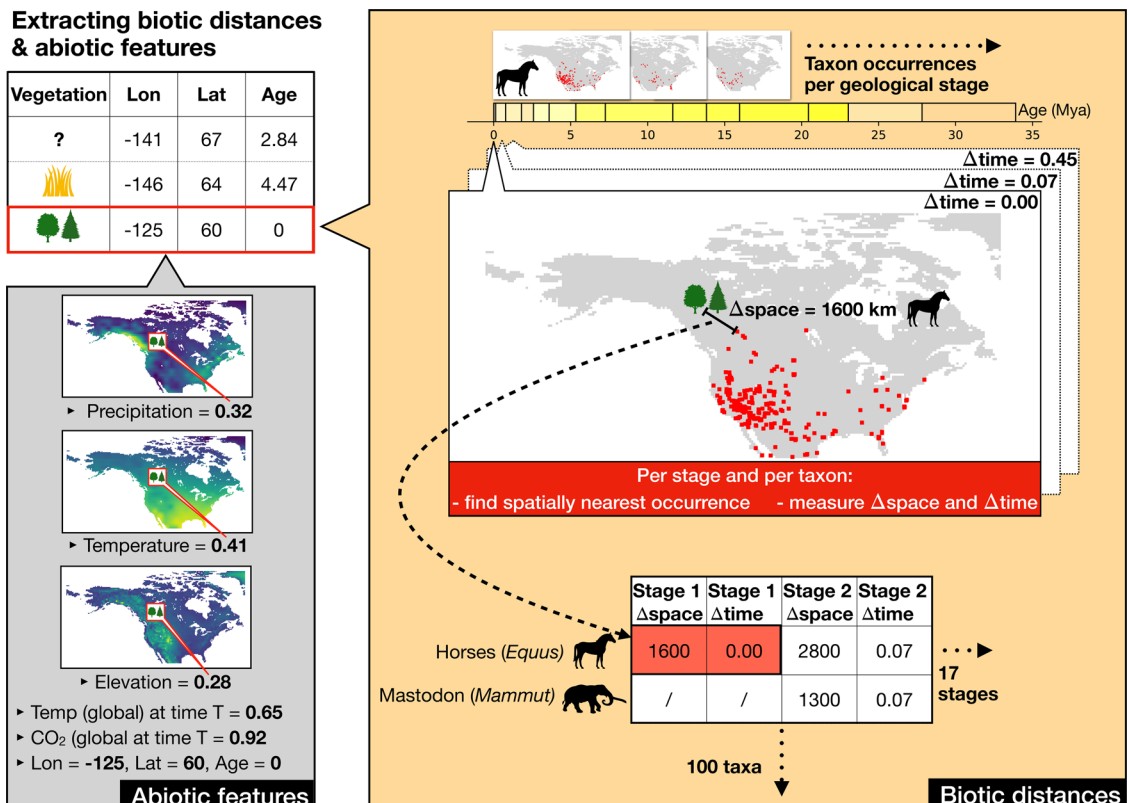

**Fig. 1 | The process of feature generation.** The workflow is shown exemplarily for one point with current vegetation information (framed in red), located at the coordinates −125, 60 (Decimal Degree System) and labeled as "closed" vegetation. Our database, compiled for this study, contains other points with current or past vegetation information, labeled as open (grass symbol) or closed (tree symbol). Once the model is trained it can be applied to estimate the vegetation interpretation for points in space and time, which are currently lacking such information (represented by the question mark). For the selected point, defined by its longitude (Lon), latitude (Lat), and age, we extract several abiotic features, reflecting climatic, geographic, and temporal variables (see box "Abiotic features"). In addition, we extract the spatial distance to the closest occurrence of each taxon in our occurrence database (see box "Biotic distances"). This is repeated for each geological stage ($n = 17$), while also extracting the temporal distance between the given point and the mid age of each geological stage. In the example the temporal distance to the nearest horse occurrence in stage 1 is 0 (see cells highlighted in red) because the vegetation point falls within this first geological stage.

"unknown", as they fall below the threshold (Fig. 3, Supplementary Fig. 6). Here, we determine a PP threshold for each of our trained models to ensure a minimum test accuracy of 90%. For the best model, a PP threshold of 0.63 was sufficient to reach an expected target accuracy of >90%, while still making vegetation inferences for 92% of the test set (Table 1), the remaining 8% being classified as uncertain.

Using the best of our trained models, we produced vegetation estimates for North America throughout the last 30 Myr in one-Myr increments. To further improve the model for predicting past vegetation, we retrained one final production model using all available paleovegetation points, including those 20% that were previously used for model evaluation. The final model was trained on all 331 paleovegetation points and 331 current vegetation points. To produce the data for the prediction task, we calculated the cell-center coordinates of all land cells in a 0.5° × 0.5° grid across the majority of the North American continent, which we delimited by a cropping window with corner points $P_1$ (Lon = −180, Lat = 25) and $P_2$ (Lon = −52, Lat = 80). We accounted for tectonic movements, transforming the grid-cell center coordinates into their equivalent paleocoordinates, using the "PALEOMAP" model of the mapast R-package[26]. From the grid-cell center coordinates, we extracted the taxon occurrence distances for feature generation, as well as all additional abiotic features, in the same manner as for the training and test data (Fig. 1).

The vegetation patterns for North America predicted by our best model show no significant evidence for the presence of open vegetation during the Oligocene epoch (23–30 Ma, Fig. 3), as the 95% HPD interval of estimated open habitat proportion during this period includes 0 (0–14.9%). Starting at 23 Ma (beginning of the Miocene) our model predicts the presence of open vegetation with certainty, as the posterior estimate of open habitat fraction significantly exceeds 0% (Fig. 4). Throughout the Miocene, these open habitats continuously expanded, forming a wide corridor roughly covering the region comprised of today's American Great Plains (Fig. 3). Already by the end of the Early Miocene (17 Ma), this ecoregion (the American Great Plains, as defined in Omernik et al.[30], see Supplementary Fig. 7) was covered by 68% (47–93%, 95% HPD) of open vegetation (Supplementary Fig. 8). Around the time of the Miocene-Pliocene transition (5 Ma) the rate of open habitat expansion across the entire North American continent began to accelerate rapidly, increasing the proportion of open vegetation from 20% (11–36%) in the Early Pliocene (4–5 Ma) to 42% (30–53%) by the beginning of the Pleistocene (2–3 Ma). During the Pliocene-Pleistocene transition, the rate of open habitat expansion reached its maximum, eventually becoming the most prominent natural vegetation type in North America today.

To determine to what degree the mammal taxa (genera) used in this study are associated with either open or closed vegetation, we evaluated for each taxon the fraction of occurrences that fall within each vegetation type, based on the vegetation predictions made by our best model, and averaged through time. The most specialized open habitat mammal genera identified by our predictions are *Cratogeomys* (gophers), *Onychomys* (grasshopper mice), *Baiomys* (pigmy mice), *Tayassu* (peccaries), *Notiosorex* (desert shrews), and *Dipodomys* (kangaroo rats), which have all of their current and fossil occurrences

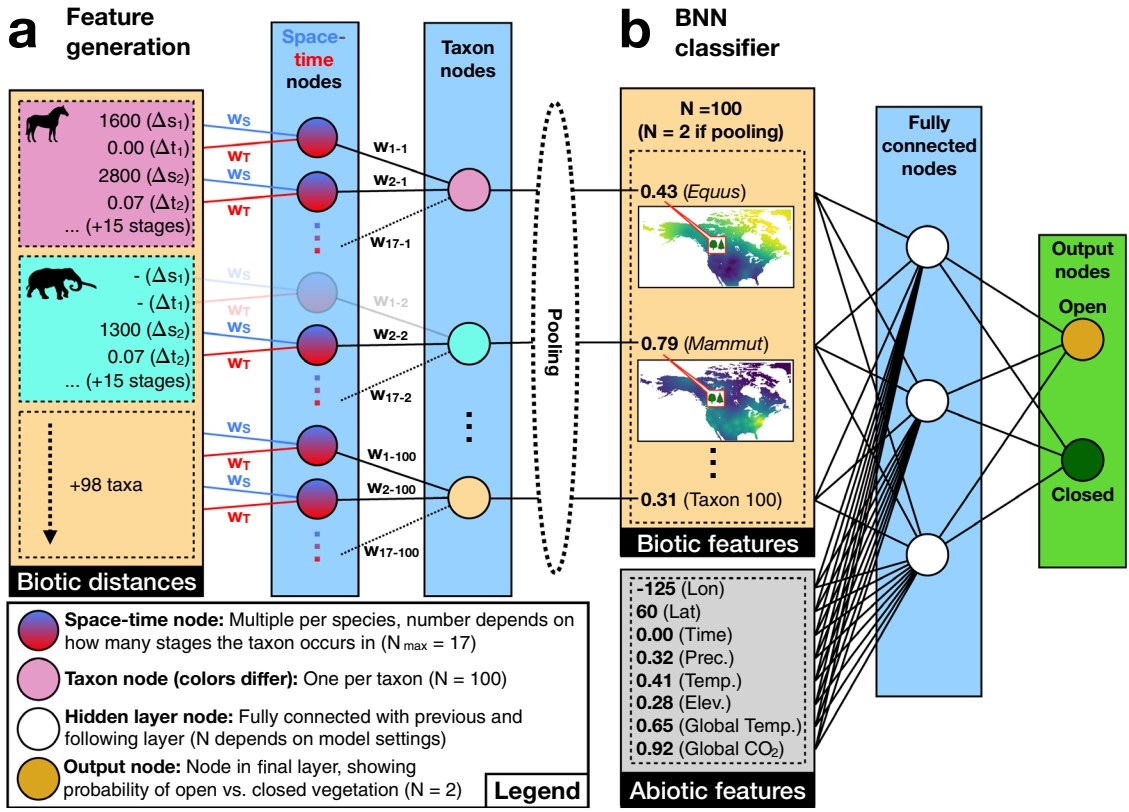

**Fig. 2 | The BNN model architecture. a** The spatial and temporal distances extracted separately for 100 mammal and plant taxa (Fig. 1), are the input of the first two hidden layers in the BNN model. During training, the BNN optimizes weights (represented by lines labeled with $w_\chi$) to reduce the multitude of spatial and temporal distance measurements into one single "proximity" value for each taxon (taxon nodes) relative to the given point in space and time. This process of feature generation is equivalent to the convolutional layers in an image classifier, reducing higher-dimensionality data into lower-dimensionality features for input into the subsequent neural network layers. In some of our tested models the resulting taxon features are pooled before being passed on to the next layer. **b** The taxon node values ("Biotic features") are then used in combination with the abiotic features as input into the fully connected BNN classifier layers. Jointly with the weights of the feature generation layers, the weights of the BNN classifier are estimated during training through MCMC sampling, to optimally map the input data to the correct output vegetation label ("open" or "closed"). Once trained, a posterior sample of the weights is stored for each model and is used to make vegetation predictions for points with unknown vegetation interpretation.

inside of grid cells with inferred open vegetation (Supplementary Table 1). These are all small-bodied taxa which are still naturally occurring in North America today (with the exception of genus *Tayassu*) and whose living species are also today associated with open habitats. On the other hand, we identified *Sciurus* (squirrels), *Dasypus* (armadillos), *Odocoileus* (white-tailed and mule deer), *Mammut* (mastodons) as the most specialized closed vegetation mammal genera (>75% of occurrences in closed vegetation). Squirrels, as well as some species of armadillos today are still associated with closed, forested habitat. Similarly, the genus *Mammut*, which included the North American mastodon, is known to have been mostly forest-dwelling until its relatively recent extinction[31]. Regarding the genus *Odocoileus*, on the other hand, both white-tailed and mule deer are known today to occur in both open and closed habitats. These species may have their origins as forest-dwelling species, having gradually adapted toward the increasing area of open grassland vegetation, a trend reflected in our data (3 Ma: 0% open habitat occurrences, 1 Ma: 21% open habitat occurrences, 0 Ma: 34% open habitat occurrences).

This demonstrates the potential of our model to estimate mammal-vegetation associations from the predictions made by the trained model, rather than having to define these a priori. However, the here-inferred habitat associations are not reflective of the importance of these taxa for our trained model, as opposed to the taxa identified by the feature importance evaluation below (Fig. 5). Further, for taxa with few fossil occurrences, the here identified habitat associations are potentially affected by incomplete sampling, as these taxa may have

occurred in both habitat types but only show occurrences in one of them in our available occurrence data.

## Sensitivity tests
We tested whether the prediction accuracy of our models improves by supplementing the paleovegetation data with current vegetation data when training the model. We found that the prediction accuracy for both paleovegetation and current vegetation increases even when adding only a small subset of the current vegetation data (331 of 11,048 grid cells) during model training (compare model 15 vs. 8, Table 1). This number of current vegetation points was chosen to match the 331 paleovegetation points that were compiled in this study. The prediction accuracy for current vegetation patterns further improves when more current vegetation data points are added during training, for example when increasing the current data 2-fold (model 16) or 5-fold (model 17), yet the prediction accuracy for paleovegetation starts to decrease. This indicates that our model utilizes some of the information that is gained from adding current vegetation information during training, but that it overfits toward the present when the number of current vegetation instances outweighs the number of paleovegetation instances. We therefore recommend for future studies using this type of model for vegetation inference to only use a subset of the available current vegetation data for training, particularly if the paleovegetation data for training is limited, to avoid overfitting towards the present vegetation pattern and towards the taxon-vegetation interactions determined for that point in time. The most

**Table 1 | Prediction accuracy of tested model configurations, with the accuracy of the best models highlighted in bold**

| Model ID | Architecture | Train instances current | Train instances paleo | Features | Pooling | Accuracy | Accuracy (paleo) | Accuracy (present) | Selected PP threshold | Predictions above PP threshold |
|---|---|---|---|---|---|---|---|---|---|---|
| 1 | 1 layer, 8 nodes | 331 | 331 | All | None | **0.887** | **0.888** | 0.876 | 0.630 | 0.921 |
| 2 | 1 layer, 8 nodes | 331 | 331 | All | Sum | 0.870 | 0.870 | 0.873 | 0.580 | 0.952 |
| 3 | 1 layer, 8 nodes | 331 | 331 | All | Max | 0.868 | 0.870 | 0.852 | 0.690 | 0.852 |
| 4 | 1 layer, 8 nodes | 331 | 331 | Only abiotic | None | 0.851 | 0.855 | 0.807 | 0.820 | 0.387 |
| 5 | 1 layer, 8 nodes | 331 | 331 | Only biotic | None | 0.883 | 0.885 | 0.861 | 0.600 | 0.943 |
| 6 | 1 layer, 8 nodes | 331 | 331 | Only biotic | Sum | 0.827 | 0.825 | 0.846 | 0.730 | 0.764 |
| 7 | 1 layer, 8 nodes | 331 | 331 | Only biotic | Max | 0.739 | 0.737 | 0.761 | 0.700 | 0.562 |
| 8 | 2 layers, 32-8 nodes | 331 | 331 | All | None | 0.874 | 0.873 | 0.879 | 0.670 | 0.924 |
| 9 | 2 layers, 32-8 nodes | 331 | 331 | All | Max | 0.871 | 0.870 | 0.882 | 0.680 | 0.879 |
| 10 | 2 layers, 32-8 nodes | 331 | 331 | All | Sum | 0.867 | 0.864 | 0.900 | 0.640 | 0.918 |
| 11 | 2 layers, 32-8 nodes | 331 | 331 | Only abiotic | None | 0.878 | 0.879 | 0.873 | 0.610 | 0.918 |
| 12 | 2 layers, 32-8 nodes | 331 | 331 | Only biotic | None | 0.868 | 0.870 | 0.852 | 0.620 | 0.921 |
| 13 | 2 layers, 32-8 nodes | 331 | 331 | Only biotic | Sum | 0.844 | 0.843 | 0.858 | 0.670 | 0.834 |
| 14 | 2 layers, 32-8 nodes | 331 | 331 | Only biotic | Max | 0.800 | 0.798 | 0.822 | 0.740 | 0.619 |
| 15 | 2 layers, 32-8 nodes | 0 | 331 | All | None | 0.849 | 0.870 | 0.644 | 0.680 | 0.906 |
| 16 | 2 layers, 32-8 nodes | 662 | 331 | All | None | 0.886 | 0.885 | 0.900 | 0.610 | 0.967 |
| 17 | 2 layers, 32-8 nodes | 1655 | 331 | All | None | **0.887** | 0.885 | **0.908** | 0.600 | 0.949 |
| 18 | 2 layers, 32-8 nodes | 331 | 0 | All | None | 0.622 | 0.595 | 0.891 | 1.000 | 0.000 |
| 19 | 2 layers, 32-8 nodes | 662 | 0 | All | None | 0.590 | 0.559 | 0.905 | 1.000 | 0.000 |
| 20 | 2 layers, 32-8 nodes | 1655 | 0 | All | None | 0.547 | 0.511 | **0.908** | 1.000 | 0.000 |

The overall accuracy of each model constitutes the weighted mean between the paleovegetation accuracy (factor 10) and the current vegetation accuracy (factor 1). The selected posterior probability (PP) threshold was chosen to reach a minimum prediction accuracy of 90% across all 331 paleovegetation points, using cross-validation. For some models this accuracy aim could not be achieved; in these cases, the posterior threshold was set to 1, leading to all vegetation predictions to be labeled as "unknown", when applying this threshold. Source data are provided as a Source data file.

suitable proportion of current vs paleovegetation points for a given vegetation prediction task should be viewed as a hyperparameter of the model, which can be fine-tuned via model testing (Table 1).

We then tested whether the addition of biotic features (taxon distances) improves the prediction accuracy, compared to models using only abiotic features, such as those used in previous studies based primarily on climate[17,18]. We find that a mixed-feature model that is trained on both biotic and abiotic features (model 1) outperforms a model that is trained on abiotic features only (model 4) by a margin of 3.6% prediction accuracy (Table 1). Further analyses of feature importance for the mixed-feature model show that several of the biotic mammal and plant features stand out as the features with a high impact on the prediction accuracy of the model, for example the genera *Ursus* (bears) and *Equus* (horses, Fig. 5). Yet, abiotic features, in particular global temperature, latitude, and time, also provide a measurable contribution toward the prediction accuracy of this model. These findings suggest that alongside previously used abiotic predictors, in our study mammal and plant fossil occurrence data capture relevant information that is used by the model to predict paleovegetation changes (Fig. 5). In combination with the outcome of our model-testing (Table 1), these results provide strong justification that models utilizing both, biotic and abiotic features should be used for the task of vegetation modeling.

Finally, we tested models in which pooling was applied to the output of the first layers of the BNN generating features from the biotic data. Specifically, we applied max-pooling and mean-pooling to the biotic features ($n = 100$) reducing them to one single faunal and one single floral feature ($n = 2$), before feeding them into the fully connected layers (Fig. 2). This approach greatly reduces the number of weights that need to be estimated, leading to faster training and better convergence of the MCMC that is used to sample the BNN weights. Both the max-pooling and the sum-pooling approach (see "Methods") led to similar results, resulting in a slight drop of about 2% in prediction accuracy of these models compared to a model not implementing

pooling (compare models 2 and 3 vs. model 1, Table 1). However, as in other deep learning models, the benefits of substantially decreasing the number of parameters may outweigh the information loss for some datasets, rendering pooling a potentially useful tool for further dimensionality reduction of the biotic features in future implementations of these models.

## Discussion

We presented a probabilistic prediction of paleovegetation and its evolution for North America, based on a deep learning model. This model extracts information from comprehensive yet underutilized data sources, including raw geographic and temporal distances to taxon occurrences, in conjunction with abiotic data such as climate, elevation, and spatiotemporal coordinates. The spatial and temporal distances that are required as input can be easily calculated for any given point in space and time, independent of its vicinity to the nearest fossil record of a given taxon, which makes our model applicable to a wide range of geographic and temporal contexts. Our approach is entirely data-driven, as it requires no previous definitions or hypotheses about mammal-plant interactions or climate tolerance limits of given vegetation types.

### Utility of BNNs

One advantage of our BNN model is that it allows for a multitude of predictors ($n = 108$ in this case). In case of the mammal and plant features, our BNN implementation is designed to automatically generate these features from the raw occurrence records (current and fossil occurrences). In classic mechanistic models (e.g., linear regression models) such high numbers of predictors are usually problematic because of the issue of collinearity of predictors, precluding these models from accurately fitting the small number of parameters towards the best unique solution. Neural networks on the other hand, including BNNs, are not affected by the collinearity of predictors, since due to the very high number of parameters in these models, many

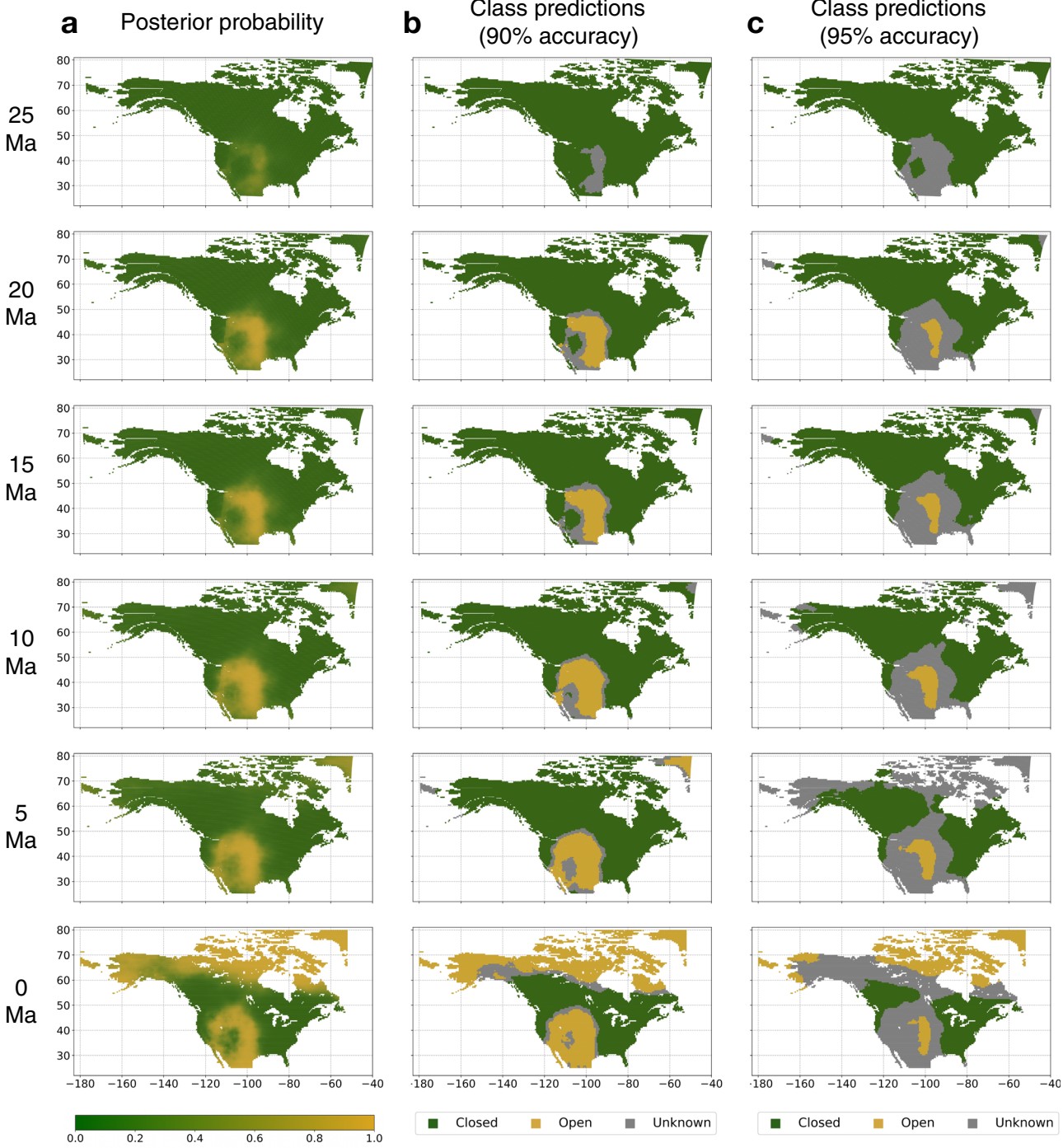

**Fig. 3 | Vegetation predictions for North America throughout the last 25 Myr.** The predictions are based on the best model resulting from our model evaluation and sensitivity tests (model 1, Table 1). Column **a** shows the posterior probability (PP) estimates for open habitat, where a PP of >0.95 (yellow) indicates strong evidence for open habitat, whereas a PP of <0.05 (green) indicates strong evidence for closed habitat. Columns **b** and **c** show categorical vegetation class predictions for our vegetation classes "open" (yellow) and "closed" (green). The class predictions are based on a PP threshold ensuring 90% prediction accuracy (**b**), and 95% prediction accuracy (**c**), respectively. The higher the applied PP threshold, the more sites will be classified as "unknown" (gray). Source data are provided as a Source data file.

solutions exist for a given problem, and fitting individual parameter estimates is not of concern[32]. This makes neural networks a suitable model to apply to even highly correlated data, such as image data which are characterized by highly correlated values between neighboring pixels. Other types of machine learning models, which have previously been used for the task of modeling vegetation (e.g., random forest models[33,34]), have been shown to be affected by collinearity of predictors, therefore often requiring an additional step of variable selection before training the final model[35]. This, however, can lead to the loss of biologically meaningful predictors and reduces the comparability between different implementations of models based on a varying selection of variables. An additional advantage of our BNN model is that it implements a direct way of quantifying the uncertainty in the class predictions. In contrast, other machine learning models

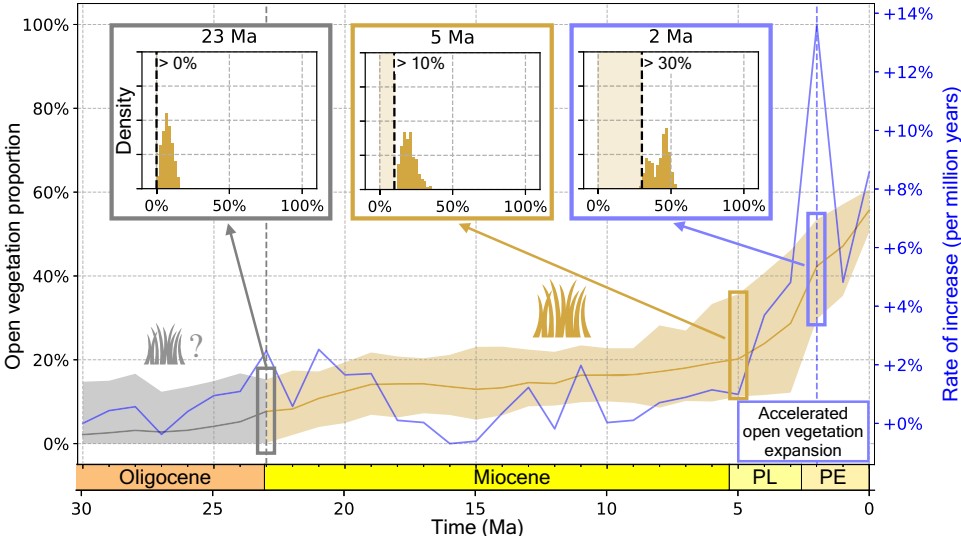

**Fig. 4 | Predicted fraction of open vegetation through time.** Fractions are calculated as the proportion of all terrestrial cells across North America predicted as open vegetation with the best model (model 1). The solid yellow line shows the mean estimates across all posterior samples, while the shaded area shows the 95% highest posterior density (HPD) interval. The blue line shows the mean rate of open habitat expansion, calculated across each preceding 1-million-year time bin. The colored bar forming the x-axis marks the geological epochs covered by our predictions, including the Pleistocene (PE), Pliocene (PL), Miocene, and Oligocene (not shown is the Holocene, from 0.01 Ma to present). The small panels show histograms of the posterior estimates of open vegetation fraction (95% HPD), marking important points in time for open vegetation evolution. These points highlight (i) 23 Ma, the earliest time where our model predicts the presence of open vegetations with confidence (>95% HPD); (ii) 5 Ma, beginning of Pliocene and the start of an acceleration in open vegetation expansion; and (iii) 2–3 Ma, beginning of Pleistocene epoch, marking the highest rate of open vegetation expansion. Source data are provided as a Source data file.

rely on approximating such uncertainties through indirect methods such as bootstrapping or other subsampling techniques of the predictor data (e.g., in random forest models[36]).

### Emergence and spread of open vegetation

While our model predictions don't exclude the presence of open vegetation prior to the Miocene, the presence of open habitat can only be inferred with confidence starting at the very beginning of the Miocene (23 Ma), based on the 95% HPD interval generated by our best model. Our paleovegetation predictions support a scenario of a comparably slow but constant rate of open habitat expansion throughout the Miocene epoch (5–23 Ma), with a recent period of accelerated open vegetation expansion starting 5 Ma (Fig. 4). This time point, marking the beginning of the Pliocene, coincides with the expansion of the more drought-adapted $C_4$ grasses at the expense of $C_3$ grasses and other plants in large parts of the North American continent, constituting a major landmark in the evolution of open grasslands[8,37]. This transition has been linked to increased aridity, which could have also led to the expansion of temperate grasslands during this period[38]. Further, our model predictions place the peak expansion rate of open vegetation at the Pliocene-Pleistocene transition (2–3 Ma). This time-period is characterized by a global drop in temperatures and the onset of glacial-interglacial cyclicity[39], putatively explaining the disappearance of forests in the northern part of the continent, and leading to an expansion of open vegetation in these areas (Fig. 3).

Our findings are in line with the scenario-based primarily on phytolith data, which places the origination and initial expansion of open habitat grasslands in the Great Plains by the earliest Miocene[9,37]. However, other types of paleovegetation data have led to the formulation of scenarios that are different to the scenario proposed here. For example, paleovegetation reconstructions based on the fossil pollen record and macrofossils place the expansion of open habitats in North America much earlier, in the late Eocene (38–34 Ma), although these open habitats were not grass-dominated[40]; on the other hand, pollen and macrofossils suggest the expansion of grassy habitats during the Middle to Late Miocene[41]. In contrast, paleovegetation

reconstructions based on paleosols suggest the presence of open habitat grasslands as early as the late Eocene[42]. This seems to imply that the interpretations of different paleovegetation data types are incompatible, leading to different conclusions. However, in this study, we trained our models with a joint dataset of phytolith, pollen, macrofossil, and paleosol data (Supplementary Data 1), as well as current vegetation information. The finding that our best models reach comparatively high prediction accuracies of around 89%, evaluated on a test set containing all data types, shows that these seemingly heterogenous data types can be successfully combined to increase the robustness of paleovegetation predictions. Unlike previous proposed scenarios of grassland evolution, our predictions are not to the same degree affected by inherent biases associated with a single vegetation data type, but instead represent a best estimation informed by the full evidence currently available.

### Outlook

We restricted our model to only two broad vegetation classes—"open" and "closed"—due to the current availability of paleovegetation data points for training. As increasingly larger and more spatially complete paleovegetation datasets are being compiled based on pollen, phytoliths, and macrofossil assemblages, this will provide sufficient training data for more detailed inferences of paleovegetation. Strategically collected data may allow predicting more detailed vegetation types, for example distinguishing between taiga, temperate and tropical forest, as well as between tundra, temperate grasslands and shrublands, and tropical steppes.

In future studies, our deep learning method could be used with high-resolution temperature and precipitation data in combination with the detailed Quaternary fossil record, to predict recent vegetation changes at spatiotemporally finer scales. For example, such models could be applied to predict vegetation changes linked to the Quaternary glacial cycles and to the human expansion and megafauna extinction (e.g., Sandom et al.[43] and Jeffers et al.[44]). Our BNN model provides an integrated framework to predict vegetation changes in deep time as well as the recent past.

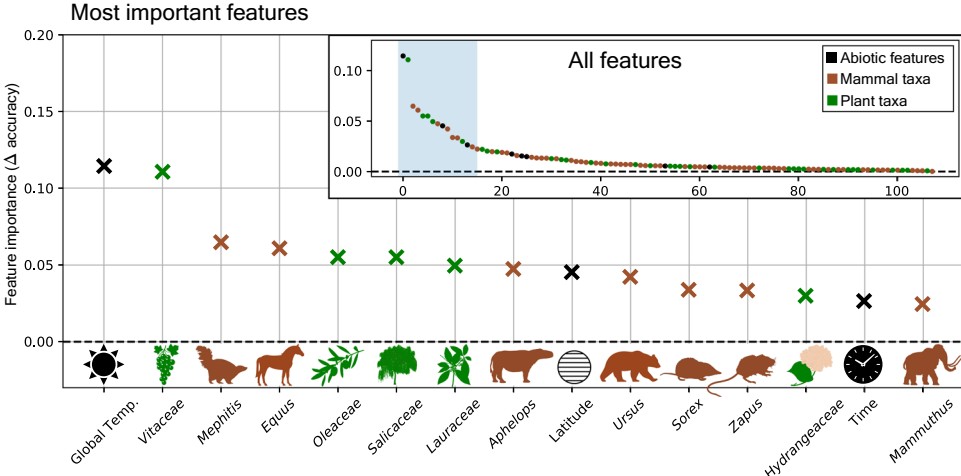

**Fig. 5 | Impact of individual features on model prediction accuracy.** The displayed delta-accuracy values (y-axis) constitute a measure of how important a given feature is for the trained model to make accurate vegetation predictions. This is determined by measuring the drop in prediction accuracy when the information content of a given feature is removed (permutation feature importance). High delta accuracy values indicate high feature importance. Points show the mean delta-accuracy of each feature across 100 randomly selected posterior BNN weight samples. The inserted panel ("All features") displays an overview of the delta-accuracy estimates for all 108 features, while the main panel displays only the most important features for the trained model. Note that the feature importance determined in this way is not an absolute measure of how important a given predictor is for the task of vegetation prediction, but rather it is an assessment of how much a given model relies on a given predictor. The identity of the most important features may change depending on the model architectures, even when based on the same data. However, the most important features identified in this manner are expected to contain relevant information for the given task, in this case for reconstructing vegetation. Source data are provided as a Source data file.

## Methods
### Data
**Spatial and temporal range.** We focused on a geographic area that is defined by a cropping window with the corner points $P_1$ (Lon = −180, Lat = 25) and $P_2$ (Lon = −52, Lat = 80), covering the majority of the North American continent (e.g., Fig. 3). We focused on the last 30 Myr, a time span encompassing most of our available sites with paleovegetation information (Supplementary Fig. 1). From the following data sources, we only selected those data points that fall within this spatiotemporal range.

Our approach described below required discretizing the input data of past vegetation labels and fossil occurrences into time-bins. For this, we chose the age boundaries of geological stages defined in the International Chronostratigraphic Chart, v2020/03[45], since these stages are expected to represent meaningful temporal units for analyzing both faunal and floral patterns. A total of 17 geological stages fell within our selected time frame of the last 30 Myr. We discretized the ages of all data points (vegetation data and fossil occurrences) that fell within a given stage by setting them to the midpoint of the respective stage.

**Paleovegetation data.** We reviewed a large body of peer-reviewed literature containing paleovegetation reconstructions and compiled a database of 331 sites with paleovegetation data for North America (Supplementary Data 1). These sites represent individual vegetation reconstructions based on fossil evidence (phytoliths, pollen, macrofossil assemblages) of distinct locations in time and space. We condensed the vegetation interpretation of the compiled vegetation data, which in many cases described specific vegetation ecosystem components, into the broader labels "open" versus "closed" vegetation. This resulted in 180 sites being labeled as closed and 151 as open, their dating rounded to the midpoint of the nearest geological stage (Supplementary Data 1). For several of these sites we found multiple vegetation reconstructions in the reviewed literature, for example when multiple sediment samples were taken from the same horizon of a given formation, belonging to the same geological stage. We treated these spatiotemporal duplicates as a single data point, excluding sites with mixed vegetation information (i.e., containing both open and closed vegetation reconstructions).

**Current vegetation data.** To supplement the limited number of paleovegetation sites, we compiled data about the current vegetation within our study area. In order to obtain current vegetation patterns, we downloaded the SYNMAP Global Potential Vegetation data[29]. As for the paleovegetation data, we collapsed the more detailed biome data into broader categories by coding the SYNMAP biome IDs < 37 as "closed" and biome IDs ≥ 37 as "open". The resolution of the SYNMAP current vegetation raster was 0.5° longitude × 0.5° latitude, which equates to a spatial resolution of ~50 × 50 km grid cells (at the equator). We extracted all current vegetation grid cells that fell within our defined cropping window, excluding all sea water cells as well as large continental lakes. This resulted in 11,048 terrestrial grid cells with current vegetation information. For these grid cells, we extracted the coordinates of the cell-center as well as the corresponding vegetation label.

The compiled paleovegetation and current vegetation points constitute the pool of vegetation information from which we sampled subsets to train our model. From here on we refer to these data points as our training instances. We trained several BNN models, using different combinations of the paleovegetation points ($n = 331$) and current vegetation points ($n = 11,048$).

**Fossil data.** We downloaded all available mammal fossil data of the last 30 Ma from the Paleobiology Database (https://paleobiodb.org/, downloaded in October 2018). We removed all entries that were not identified to species level, as well as all spatiotemporal duplicates. In several cases, the fossil data downloaded from the major databases contained minor spelling inconsistencies in the genus names and species epithets. To correct these misspellings, which can lead to an overestimation of the number of genera and species in the dataset, we used the algorithm implemented in the PyRate package[46], which automatically identifies common typos in scientific names. Finally, we removed all aquatic families from the dataset (dugongs, pinnipeds, and whales).

For each fossil occurrence, we determined the mean age of the respective stratigraphic age interval. We reduced the taxonomic resolution of the mammal data to genus-level with the main purpose to reduce the number of taxa, while increasing the spatial and temporal extent of each taxon as well as to avoid taxonomic biases such as oversplitting or lumping of species in different genera, depending on taxonomic authority. These potential taxonomic biases are expected to have a smaller impact on genus level compared to species level. To further reduce the number of taxa to only the most informative ones, we only kept genera that were present in more than half of the geological stages covered in this study, based on the first and last occurrence date of each genus in the fossil record (assumed presence in at least 9 of 17 stages). This resulted in 65 selected mammal genera (Supplementary Table 1). While the model can potentially handle any number of taxa, taxa with occurrences spanning multiple locations and time bins are expected to be most informative in our supervised learning approach.

As an addition to the mammal fossil data, we compiled a large dataset of plant macrofossils from the Cenozoic Angiosperm database[24]. Due to the sparse fossil record of plants with a taxonomic resolution of species or genus level, we decided to reduce the taxonomic resolution of the plant fossil data to family level. As with the mammal fossil data, we took the mean age of the stratigraphic age interval of each fossil occurrence and only selected plant families that were present in North America during at least 9 of the 17 geological stages. This resulted in 35 selected plant families (Supplementary Table 1). The final fossil data, consisting of the selected mammal and plant taxa ($n = 100$), amounted to a total of 5514 fossil occurrences (4770 mammal and 744 plant fossils, Supplementary Data 2).

**Current occurrences.** To complement the occurrence data extracted from the fossil record, we extracted current occurrences for all selected taxa from the Global Biodiversity Information Facility (GBIF, www.gbif.org, accessed in September 2019). For all mammal genera we downloaded the data through the R-package *rgbif*[47], only allowing human observations (as opposed to, e.g., machine observations or fossil data) and restricting the search to North American occurrences (Canada, Mexico, or USA), using the following command:

*occ_search(taxonKey=taxon_id, return="data", hasCoordinate=TRUE, country=c('US','CA','MX'), basisOfRecord='HUMAN_OBSERVATION')*

Due to the large data volumes for the selected plant families, which result in very long waiting times and occasional time-out errors when using the rgbif package, we instead downloaded the current occurrences of the selected plant families directly from the GBIF online interface (download https://doi.org/10.15468/dl.nxuyg8).

After filtering these occurrences to exactly match the cropping window defined in this study (see above), this resulted in a total of 1,299,782 current occurrences for the selected extant mammal and plant taxa (109,027 mammal and 1,190,755 plant occurrences, Supplementary Data 2). Finally, all fossil and current occurrences of the selected taxa were merged into one data-frame and jointly treated as occurrence data, independently of the data origin as fossil or GBIF observation. For all further steps, we only selected those occurrences that fell within the cropping window defined as described above.

While the current distribution of taxa—and in effect their recorded spatial occurrences—are affected by human impact (a bias that is not present in the fossil occurrence data), we are assuming here that these current occurrences are still informative about a taxon's habitat preference. This assumption holds true, unless there is reason to assume that taxa completely shift their habitat preference from open to closed habitat or vice versa, due to human impact. For the purpose of this study, we don't expect this assumption to be violated. Only if this assumption was violated for a substantial number of taxa would we expect this potential bias to affect our model predictions.

**Climate and elevation models.** The paleoprecipitation and temperature data were reconstructed based on global climate raster data with a spatial resolution of 1° longitude × 1° latitude (raw data provided by Christopher Scotese). These rasters derive from the PALEOMAP Project, which has produced paleogeographic maps at 5-Myr intervals[25] and has assembled related precipitation and temperature data based on the HadleyCM3 paleoclimate simulations[48]. Similarly, we downloaded global elevation rasters through time, generated by Scotese and Wright[49]. Because the paleoclimate and elevation estimates are only available in 5 Myr intervals, we linearly interpolated the values into 1 Myr year intervals to reach higher temporal resolution. Since no directly measurable and spatially explicit and complete data exists to inform our models about climate and elevation through deep time, we apply these estimates—which themselves have been generated through modeling—as part of the input data for our models. To test to what extent potential biases or errors in these modeled data may affect our model predictions, we added increasing levels of noise to these data before making predictions with our models. For each of these predictors (precipitation, temperature, and elevation) we randomly resampled values for each grid cell from a uniform distribution ranging between ±10%, 20%, and 50% of the original value. These modified data were then used in combination with all other features to make vegetation predictions, to quantify how such uncertainties in the data affect our model predictions. This had no detectable effect on our vegetation predictions, as can be seen based on the produced vegetation maps for the 50% perturbated climate and precipitation grids (Supplementary Figs. 9 and 10).

As additional predictors, we downloaded estimates of mean global temperature that are based on oxygen isotope data[27], and mean global atmospheric $CO_2$ concentration estimates based on carbon isotope data from fossil soils and stomatal pore density of fossilized leaves[50]. In theory, there are many other predictors that would be useful for the task of vegetation prediction, such as seasonal climatic variables and fluctuations of different elements in the atmospheric composition. However, the limitation is usually that these predictors are not available throughout the whole time frame covered in this study (last 30 Myr), particularly not in a spatially explicit manner as spatial grids. Future studies may be able to compile such data throughout deep time (based on measurements or modeled data) and be able to apply them as additional predictors in models similar to the ones presented here.

## Feature generation

An essential element of applying neural networks is the process of feature generation, which describes the transformation of the raw data into numerical features that can be fed into the neural network. Each input data point, which is commonly referred to as an instance, consists of a list of associated feature values. In our case, the training instances consist of specific points in space-time with available vegetation information, and the associated features contain the information about nearby occurrences of the selected taxa (biotic features), as well as other information about climate, geography, and time (abiotic features), in relation to the given point.

**Biotic features.** For a given instance (vegetation point), defined by its spatial and temporal coordinates, we extracted the geographic distance between this instance and the closest occurrence of each taxon, and we did so for each geological stage (Fig. 1). To calculate these distances, we transformed all geographic data into the Albers equal area projection and then calculated the distance between a given pair of coordinates in this projection. If a taxon was present in all stages, this resulted in 17 geographic distances extracted for this taxon, one for each stage. These spatial distances were calculated using the current coordinates (instead of the paleocoordinates) of each point, assuming that the relative spatial distance between any two given

points within North America is not affected (or negligibly so) by continental movements during the last 30 Myr, although their absolute coordinate values have changed through time.

In addition, we extracted the temporal distances between the selected taxon-occurrences and the given vegetation point, by measuring the difference between the age of the training instance and the midpoint of the geological stage of a given taxon occurrence. This resulted in $N$ pairs of geographic and temporal distances to each taxon, where $N$ is the number of stages this taxon occurred in. We designed our BNN model to estimate parameters to summarize the spatial and temporal distances of the selected occurrences of each taxon into one taxon-specific feature value, representing a measure of general "proximity" of each taxon, which we explain in more detail below (Fig. 2).

**Abiotic features.** In addition to the biotic features, we extracted the temperature, precipitation, and elevation associated with the space-time coordinates of a given instance. For this step we transformed the coordinates of each given vegetation label into the equivalent paleo-coordinates at the time of the record, using the "PALEOMAP" model of the *mapast* R-package[26]. We extracted the modeled temperature, precipitation, and elevation of these paleocoordinates from the ras-terized climate and elevation data[25] as three separate features. In addition, we extracted the mean global temperature and the average atmospheric $CO_2$ concentration at the given time point. Finally, we added the absolute paleocoordinates (longitude and latitude) as well as the age of the vegetation point as three additional features.

Our neural network was trained on a total of 100 biotic features (one for each selected taxon), 4 climatic features, 1 elevation feature, and 3 spatiotemporal features, resulting in a total of 108 features for each instance.

To avoid potential biases based on the absolute values of given features, we scaled all features to a range between 0 and 1. The rescaling was done jointly for all training and prediction instances, in order to avoid differences in rescaling-factors between features in the training instances and those in the prediction instances.

### Selecting training and test data

For the training of our neural network we had a total of 11,379 points with vegetation information available, consisting of 331 paleovegetation points and 11,048 current vegetation points. To test whether the larger number of current vegetation instances might bias our past vegetation predictions, we explored different combinations of paleo-vegetation and current vegetation instances during training of the model (Table 1).

To evaluate the prediction accuracy of our trained models, we performed a five-fold cross-validation, training each of the five cross-validation models on 80% of the available instances, while sparing the remaining 20% as a test set. The instances for each cross-validation fold were selected ensuring the same proportion of paleovegetation instances and current instances in each cross-validation fold. We then determined the prediction accuracy of the model as the average test set prediction accuracy across all 5 cross-validation folds, which we determined separately for all paleovegetation instances and all current instances. The final prediction accuracy of each model was then determined as the weighted mean between the paleovegetation prediction accuracy and the current vegetation prediction accuracy of the model, weighing the paleovegetation component ten times higher, as it represents the accuracy across ten geological stages that are covered by our paleovegetation data (Supplementary Fig. 1), while the current data only represent a single geological stage.

### Neural network configuration

We developed a BNN classification model that maps raw spatial and temporal distances of selected taxon occurrences (fossil or current) to a set of vegetation classes. These distance features can be complemented by any set of additional features, such as the abiotic features used in this study. The BNN model consists of multiple hidden layers generating a numerical representation of the features in multi-dimensional space, as well as an output layer that maps the nodes of the last hidden layer to the output classes, in this case open and closed habitats. Given the flexibility of our model and the fact that it is based on absolute distance measures, it may be applied to any vegetation prediction task, independently of the spatial and temporal scale of the data.

The first two hidden layers are only applied to the taxon distance features, not to the additional abiotic features. In these layers, the raw spatial and temporal occurrence distances are combined into a single value per taxon, which represents a measure of proximity of each taxon to a given input instance.

The raw distances are provided in pairs of one spatial and one temporal distance measurement, both associated with a specific occurrence of a taxon. We indicate with $\Delta s_{ij}$ and $\Delta t_{ij}$ the spatial and temporal distances for a species $i \in \{1, \ldots, I\}$ at a geological stage $j \in \{1, \ldots, J\}$ (Fig. 1). These are used as input in a first hidden layer (Eq. 1) of a sparse neural network with parameter sharing resulting in one node for each species and geological time:

$$h_{ij}^{(1)} = g\left(w_s \Delta s_{ij} + w_t \Delta t_{ij}\right) \tag{1}$$

where $w_s$ and $w_t$ are weights associated with space and time distances, respectively, shared among all species and occurrences and $g(\cdot)$ is the *swish*[51] activation function (Eq. 2):

$$\text{swish}(x) = x \times (1 + \exp(-x))^{-1} \tag{2}$$

The *swish* activation function was used after each hidden layer in the model. To reduce the number of estimated parameters for better convergence, the space and time weights are shared among all occurrences under the assumption that the relative importance of space and time in determining the proximity of a given occurrence is expected to be the same for all occurrences of different taxa.

After combining spatial and temporal distances into one spatio-temporal distance value in this way, we estimate specific taxon-weights for each taxon and geological stage, which are then used to collapse the multiple spatiotemporal distances across different geological stages into one single feature value for each taxon. This happens in the second hidden layer (Eq. 3):

$$h_i^{(2)} = g\left(\sum_{j=1}^{J} h_{ij}^{(1)} W_{ij}^{(2)}\right) \tag{3}$$

where $W^{(2)}$ is a matrix of weights associated with each geological stage $j$ specific to taxon $i$. The second hidden layer $h^{(2)}$ thus includes one node for each species, which provides the input, along with additional abiotic features $f$, to a fully connected neural network. Depending on the chosen pooling strategy, these taxon feature values are either fed as individual features into the next layer (no pooling) or are summarized into one faunal and one floral feature, by either extracting the maximum output value from layer $h^{(2)}$ (max-pooling) or by summing all output values (sum-pooling) across all mammal and plant taxa, respectively.

Following the initial two layers, the taxon-features ($h^{(2)}$, $n = 100$ or $n = 2$, depending on pooling strategy) are fed together with the additional abiotic features ($f$, $n = 8$) into a fully connected neural network and eventually mapped to the binary vegetation classes in the output layer (Fig. 2b). Given a set of input features $x = \{h^{(2)}, f\}$ of size $M$ the next hidden layer (Eq. 4) with $n$ nodes is obtained through:

$$h_n^{(3)} = g\left(\sum_{m=1}^{M} x_m W_{mn}^{(3)}\right) \tag{4}$$

where $W^{(3)}$ is a matrix of $M \times n$ weights. Finally, the output of the neural network (Eq. 5) is binary and quantifies the probability associated with each class (closed and open habitats):

$$y_o = \sigma\left(\textstyle\sum_n h_n^{(3)} W_{no}^{(4)}\right) \qquad (5)$$

where $o = 2$, $W^{(4)}$ is a matrix of $n \times 2$ weights, and $\sigma(\cdot)$ is the *softmax*[52] function (Eq. 6):

$$\sigma(x_k) = \frac{\exp(x_k)}{\sum_o \exp(x_o)} \qquad (6)$$

We tested different network configurations in terms of number of layers and nodes per layer, different pooling strategies, as well as different combinations of training features and instances, and selected the best model based on the highest test set prediction accuracy (Table 1).

The parameters of the model ($w_s, w_t, W^{(2)}, W^{(3)}, W^{(4)}$) were jointly estimated using a Metropolis Hastings Markov Chain Monte Carlo (MCMC) algorithm[53]. During training, all weights of the model are initially drawn randomly from a normal distribution centered in 0 and are then updated via MCMC sampling. We used a standard normal distribution as prior on all weights (parameters of the model). During model testing we ran an MCMC chain for 200,000 generations for each cross-validation replicate, sampling every 200 iterations. We selected the best model based on the highest prediction accuracy, and then trained a final production model with these best model settings using all available instances (no test set) for 400,000 additional MCMC generations, departing from the parameter values estimated during cross-validation.

Our BNN implementation allows not only to estimate the most probable vegetation label for a given point in time and space, but also to calculate the posterior probability of this label, providing an inherent measure of uncertainty. We calculated the posterior probability of each class label for a given instance as the mean class probability across all posterior samples. This ability makes BNNs an attractive alternative to regular neural network algorithms, which allow no such uncertainty modeling, although analogous approximations exist, such as Monte Carlo dropout[54].

### Feature importance

To determine the relative importance of each feature used in our model, we applied the method of permutation feature importance (sensu Breiman[55]). In this approach, the values of a given feature are randomly shuffled across all instances of the test or training set. This process masks any existing information that lies within the data of a given feature. The class labels for all instances are then predicted using the modified feature matrix. The resulting prediction accuracy is then compared with that of the original feature matrix and the difference between these accuracies ($\Delta acc$) is interpreted as a measure of relative importance of the shuffled feature for the classification task. We repeated this process for each feature column in our feature matrix ($n = 108$), using the complete training set, and ranked the features based on their $\Delta acc$ values (Fig. 5).

### Predicting vegetation labels

To produce continuous vegetation maps across North America, we constructed a 0.5° × 0.5° grid across the cropping window defined in this study and extracted the coordinates of the cell-center for each grid cell ($n = 11,731$). For these points, we extracted spatiotemporal taxon distances and abiotic features in the same manner as for the training instances. We repeated this process in 1 Myr steps starting in the present ($t = 0$) throughout the last 30 Myr ($t = 30$), producing 31 feature-datasets of North America through time, considering tectonic movement (*mapast*[26]). Based on the BNN weights sampled during training by the MCMC (excl. burn-in) we determined the posterior probabilities of each vegetation label for each given point (Fig. 3).

To produce more spatially explicit subsets of the North America grid, we downloaded shape files delineating the ecoregions of North America (Level 1 ecoregions[30], downloaded from https://www.epa.gov/eco-research/ecoregions-north-america). We identified all grid cells that fall within each ecoregion and extracted the vegetation predictions for these cells, to track the spread of open vegetation in each of these ecoregions separately.

### Reporting summary

Further information on research design is available in the Nature Research Reporting Summary linked to this article.

## Data availability

The Supplementary Information accompanying this manuscript contains Supplementary Discussion, Supplementary Figures 1–10, and Supplementary Table 1. In addition, Supplementary Data 1 and 2 are available in the Zenodo repository https://doi.org/10.5281/zenodo.6492100. The repository also contains all datasets analyzed and generated during the current study, as well as source data for all figures and tables. Specific data sources used in this study were: (i) paleovegetation reconstructions from peer-reviewed literature (see Supplementary Data 1); (ii) current vegetation information from SYNMAP Global Potential Vegetation data (https://databasin.org/datasets/1l2a942ec4294e5284e63d5e6bf14b29/); (iii) mammal fossil data from Paleobiology Database (https://paleobiodb.org/, see Supplementary Data 2); (iv) plant fossil data from Cenozoic Angiosperm database (https://doi.org/10.1086/685388, see Supplementary Data 2); (v) current taxon occurrences from GBIF (download https://doi.org/10.15468/dl.nxuyg8); (vi) elevation rasters through time (https://zenodo.org/record/5460860); (vii) paleotemperature and paleovegetation data through time (https://doi.org/10.1146/annurev-earth-081320-064052). Restrictions apply to the availability of these data, which were used under license for the current study, and so are not publicly available. Data are, however, available from the authors upon reasonable request and with permission of Christopher Scotese (cscotese@gmail.com).

## Code availability

All code used in this study, as well as a full data tutorial and installation instructions for training BNN models and predicting vegetation through time, are available on the project's GitHub repository (https://github.com/tandermann/paleovegetation[56]). The main BNN functionalities can be loaded as a stand-alone and open-source Python package, which is available on GitHub (https://github.com/dsilvestro/npBNN, v0.1.12), allowing the application of the described BNN approach for any classification or regression task, not only restricted to the task of vegetation prediction.

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

## Acknowledgements

We thank all paleontologists who have produced and made publicly available the paleovegetation data and fossil occurrence data necessary for training the models presented in this study, Christopher Scotese for providing climate models for paleotemperature and paleoprecipitation, Thomas Alan Neubauer for advice on the paleoclimate and elevation models, Juan Carrillo for providing feedback on the mammal fossil data. T.A. received financial support from the SciLifeLab & Wallenberg Data Driven Life Science Program (grant: KAW 2020.0239). T.A. and D.S. received funding from the Swedish Research Council (2019-04739). D.S. received funding from the Swiss National Science Foundation (PCEFP3_187012). A.A. acknowledges financial support from the Swedish Research Council (2019-05191), the Swedish Foundation for Strategic Research (FFL15-0196), and the Royal Botanic Gardens, Kew. C.A.E.S. acknowledges the United States National Science Foundation (EAR-1253713). All computations were carried out on the Kebnekaise computing cluster, as part of the High Performance Computing Center North (HPC2N), which is funded by the Swedish National Infrastructure for Computing (SNIC), as well as the Kempe Foundations and the Knut and Alice Wallenberg Foundation.

## Author contributions

T.A., A.A., and D.S. contributed to conception and design of the study. C.S. compiled and revised the paleovegetation data. T.A. compiled all other data, wrote the code with contributions from D.S., ran all analyses, and wrote the first draft of the manuscript with contributions from all authors. All authors contributed to the article and approved the submitted version.

## Funding

## Competing interests

The authors declare no competing interests.
