## [Peer Review File · Nature Communications]

The origin and evolution of open habitats in North America inferred by deep learning modelsReviewers' Comments:

Reviewer #1:

Remarks to the Author:

Andermann et al. study the evolution of terrestrial biomes in North America since 30 million years ago (Ma) using a Bayesian deep learning model trained by some available biotic and abiotic information. In particular, it focuses on the open habitat transition, and shows the origin and expansion of grasslands in North America, which first developed around 23 Ma and reached 50% of extent after 7 Ma. The manuscript is clearly written and easy to understand its message. The results are interesting and are fully explored with additional sensitivity experiments. The authors provide a high (spatial) resolution, million-year resolved dataset of open and closed biome types that can be used for future intercomparison studies.

However, I have some major concerns about the methodology and the results, which I think need to be addressed before the paper can be published.

1. The application of the BNN model is not well described and does not seem to have been very well implemented. The key reference (or equations) is missing for the model. How the model treats dependent (e.g. temperature and precipitation) or independent predictors (e.g. longitude and mammalian fossil type), or maybe conflict predictors are not explained. There is no mention of the issue of colinearity in the selection of the predictors, and whether this was assessed. Some important predictors are not included (see my next point). For a general reader, a brief discussion on the advantage of BNN compared to other ML methods such as random forest (Hengl et al., 2018) will also be interesting.

2. I think the greatest problem the current experimental protocol suffers is the lack of some important predictors. For instance, the elevation/landscape, the seasonal features in climate variables, and, considering the deep time, the atmospheric composition.

The uplift of cascades can cause higher elevation cooling that preserves snowpack, and also orographic precipitation and perennial rivers. These factors (by preserving forests) can further affect grassland expansion near the mountain slopes in the midwestern North America (see an overview in Kukla et al., 2021), and in these regions are the boundaries between the open and closed habitats and have the highest uncertainties. A parameter describing elevation/landscape should be taken into account.

I would also say that annual mean temperature and precipitation might not be the crucial predictor for vegetation, and it would be more reasonable to look at the real controls on vegetation, e.g., coldest month temperature, growing season warmth and plant-available moisture. So, my suggestion would be to include summer and winter temperature and precipitation as predictors.

Also, given the time horizons explored in this study, that vegetation can adapt to different atmospheric compositions, particularly the stomata density and photosynthesis/respiration acclimation to varying CO₂ levels, which can potentially modulate the vegetation type. So pCO₂ can also be included as a predictor, or be tested.

3. The authors discussed the open habitat transition in the whole North America, and a corresponding 50% threshold suggests a relatively recent expansion of grasslands. However, if the authors use smaller subregions instead, such as the midwestern North America, the timing of the transition (e.g. keep the 50% threshold for this subregion) changes and can lead to unsurprising results. Perhaps while keeping the main conclusion for the whole North America, the authors can add predicted fraction (similar to Fig. 3) also for these subregions and have a more comprehensive comparison with previous studies focusing on smaller regions.

4. The uncertainties in the input are completely ignored in the study. I wouldn't regard climate fields from ref24 (is that the right source by the way? I cannot find the 1 deg precipitation and temperature dataset from this paper) as 'reliable' predictors without any tests as they were generated by a model.

Linear interpolation of the climate fields from 5 Myr to 1 Myr intervals also seems sloppy. It would be useful to know how much the biases in climate fields can lead to uncertainties in predicted vegetation fraction, say, by perturbing the climate variables systematically (e.g. Lindgren et al., 2021) by $\pm 10\%$, $\pm 20\%$? Maybe the authors have an alternative way of quantifying this, and, if so, it would be useful to include in your results. Additionally, I think the authors should be cautious to use 'reconstruction' (e.g., L259, L275) to describe the probabilistic model results.

Minor points

L65 Please provide key references for the BNN you used.

L85 How does the mean global temperature from ref26 compare with the global mean of annual temperature from ref24?

Fig. 1 This figure is useful and contains a lot of information. Please reorganize it to make it easier to read. Please also add more details, for instance, the colors of taxon nodes, fully connected nodes, and output nodes.

L113 An alternative for the SYNMAP global potential vegetation data is the pre-industrial vegetation reconstruction, e.g., LandCover 6k biome reconstructions (Harrison et al., 2020, Figure 7)

L155 Is the 20% randomly selected?

Fig. 3 Maybe 5 Myr interval should be highlighted, so the readers are aware of the interpolation of climate fields.

L230 & Table 1 I am confused here. Is this outperformance reflected by the table?

Fig. 4 how to interpret the values of importance? What does it mean when it's negative? The importance of the 3rd to 15th features seems very comparable.

L325 Each stage has a long span, and I cannot find how the mean age is calculated from max and min ages (e.g. in supplementary data S2).

L328 Some of the vegetation types are underrepresented in the data, e.g. the desert. If there is desert (not in Table S1 though), did you classify it as open habitat?

L391 Just like the vegetation cover is affected by human land alterations (L114), the occurrences of biodiversity are also changed by human activities. But there are no 'potential' biodiversity maps.

L410 Why not also interpolate climatic predictors to midpoint of each geological stage?

refs

Harrison, S.P. et al. (2020). Development and testing scenarios for implementing land use and land cover changes during the Holocene in Earth system model experiments. *Geoscientific Model Development*, 13(2), 805-824.

Hengl, T., Walsh, M. G., Sanderman, J., Wheeler, I., Harrison, S. P., & Prentice, I. C. (2018). Global mapping of potential natural vegetation: an assessment of machine learning algorithms for estimating land potential. *PeerJ*, 6, e5457.

Lindgren, A., Lu, Z., Zhang, Q., & Hugelius, G. (2021). Reconstructing past global vegetation with Random Forest Machine Learning, sacrificing the dynamic response for robust results. *Journal of*

Advances in Modeling Earth Systems, 13(2), e2020MS002200.

Kukla Tyler et al. (2021). <https://doi.org/10.1002/essoar.10507866.1>

Reviewer #2:

Remarks to the Author:

This is an exciting study that addresses a classic scientific question – when and where did open vegetation in North America began to emerge over the last 30 million years – through the combination of deep learning models and a compilation of paleoecological data from across an array of paleobiological and biological data resources. The paper is a technical tour de force, presenting in detail a Bayesian Neural Network (BNN) to infer past vegetation distributions from mammal data. The assembly of paleovegetation and mammalian fossil occurrence data from across a range of paleodata resources is also impressive. The compilation of paleovegetation data is particularly valuable.

The description of the BNN structure and results are clear and thorough. Note that I am not a Bayesian statistician myself, but I have collaborated with them. So I cannot give a deep assessment of the modeling approach, but I read the model description closely and everything made sense to me.

In some ways, the key question posed by this paper (did open vegetation first form 28-23 Ma vs. 23-16 Ma?) isn't definitively answered, but this paper now clearly establishes where we can confidently see the establishment of open vegetation (after 23 Ma) and where there is persistent uncertainty (before 23 Ma). And the maps of open and forested vegetation over the last 30 Ma (Fig. 2) are fantastic. The paper's major advances are 1) bringing the most data to bear yet on this question and 2) its innovative methodological approach towards integrate many different lines of paleoecological evidence into one unified inferential synthesis.

I have a couple of moderate comments about the data analysis and visualizations. The code needs to be made openly available and the data need better documentation – see below for more information.

1. Using Geographic distance to calculate spatiotemporal distance is not great (L421-423), especially for a dataset that spans so many latitudes, where the km per degree longitude varies by a factor of ~50%. It would be better to reproject all spatial coordinates onto e.g. Albers equal area grid and then do the distance calculations. It's a pretty important issue, because these spatiotemporal distances are the fundamental input used by the BNN. I would personally recommend recalculating distances and rerunning of the model. However, I recognize this would be a lot of work, so the other non-preferred but possible option would be to simply note in the text that this is a known issue.
2. L433-436 (& Fig 1): The use of the BNN to come up with a combined weighting of spatial and temporal distance is very interesting. I'd be quite interested to see a table or other summary of those weights, to get a sense of how much relative weight the model gives to spatial distance vs. temporal distance.
 - a. L486-488: Now I see that all values are collapsed to a single spatial weight and single temporal weight. I'd be curious to see those.
3. Given that this paper is primarily presenting a new modeling approach, the code needs to be made open and available, both for reproducibility purposes and to help advance the field. Deposition in an open GitHub repository would be sufficient, with a branch version representing the state of the code at the time of the ms.
4. More documentation of the data is needed, so that the reader can better judge the strength of the data underlying these findings, and for reproducibility.
 - a. Add maps in SuppInfo that show the distribution of a) paleovegetation data points and b) mammal/plant occurrence data points for each time period.
 - b. Possibly could add a second set of maps that code the paleovegetation data points by dataset type (pollen/spores, macrofossils, etc.)

- c. Upon acceptance of ms., Tables S1 and S2 should be moved out of their current Google Drive and a current permanent archive such as Dryad.
- d. Supplementary Figure S1: Color-code the histogram so that for each time period, the proportion of open and forested vegetation is shown.
- 5. Supplementary Table S1: This is impressively detailed! A great resource. However, many columns in the table are cryptic or incomplete, so the table needs an explanatory legend or readme file. Specific comments for each column:
 - a. A/B: Lat/Lon: Specify units (DD)
 - b. C/D: Age Max/Min: Specify units: Ma?
 - c. E/F: Author name/Title/Year: OK but a DOI would be even better. Citation indexers would have a hard time scanning this table and retrieving citation usage.
 - d. 'Entered by': Author codes (T, K, ...) are cryptic. Either show names in full or provide a lookup table somewhere.
 - e. 'Reference #:': what does this store? Seems to hold book titles. What does an empty field indicate?
 - f. 'Locality name': Is this a spatial place name or a geostatigraphic formation name?
 - g. Depth/Location: Units? Some fields have units and some just have unitless numbers.
 - h. For Columns N onwards, which of these represent information drawn straight from the original publication and when is it based on interpretation of the synthesis authors?
 - i. R: Title ('Colors match') doesn't match field content, which is country names.
 - j. What's the difference between Columns AR & AS, which both indicate C3 vs C4?
- 6. Supplementary Table S2: Needs more information.
 - a. Where do all these fossil taxa come from? They are unprovenanced. Need database name, and original publication. Original dataset unique ID would also be helpful.
 - b. Add a column indicating whether a given taxon name is a mammal or plant (most readers will know, but will help clarify)
 - c. Columns B:G need units

LINE-BY-LINE COMMENTS

L8: already->by

L28: Here and elsewhere: the reference to grasslands as the most extensive terrestrial biome 'today' overlooks the widespread anthropogenic conversion to agriculture. Need to either note agricultural conversion or shift 'today' back at least a few centuries.

L35: Unclear antecedent for 'their'

L64-69: Clarify in this opening description that you are training the BNN against independent vegetation datasets, both paleo and modern. Took me a deep read before I understood this basic structure of the modeling approach.

L69-75: Also add to this opening paragraph 1-2 sentences that state that you train this model against newly built paleovegetation (N=331) and fossil mammal/plant occurrences (N=7500). This data compilation is as impressive as the model itself, and is a major contribution of this paper.

L196 hyphenate 'here-inferred'

L278-291: The discussion here could use sharpening: Should this paper's findings be interpreted as 'no open vegetation during the Late Eocene (i.e. nothing before 23 Ma, Fig. 3)' or 'not enough data to assess whether there were open habitats during the Late Eocene'?

L292-299: Might note here that the effect of temperature on open vegetation did not emerge until the Pleistocene (1Ma), when Pliocene-Pleistocene cooling resulted in a large area of open tundra across the Arctic (Fig. 1Ma). So the temperature effect is there but later/secondary.

L340: Was this random choice of open vs. closed change for these three sites allowed to vary among analytical runs, or was it chosen once and then kept fixed for all analyses?

We thank the editor and the two reviewers for the thorough and insightful assessment of our manuscript. We carefully addressed all comments and suggestions, which has considerably improved the clarity and quality of the revised manuscript. Please find below our point-by-point response to all comments. Our replies to the reviewers' comments are marked with "Response:" at the beginning of the paragraph and are colored in blue. All line numbers refer to the manuscript version with marked track-changes, which is attached at the end of this document.

Reviewer #1 (Remarks to the Author):

Andermann et al. study the evolution of terrestrial biomes in North America since 30 million years ago (Ma) using a Bayesian deep learning model trained by some available biotic and abiotic information. In particular, it focuses on the open habitat transition, and shows the origin and expansion of grasslands in North America, which first developed around 23 Ma and reached 50% of extent after 7 Ma. The manuscript is clearly written and easy to understand its message. The results are interesting and are fully explored with additional sensitivity experiments. The authors provide a high (spatial) resolution, million-year resolved dataset of open and closed biome types that can be used for future intercomparison studies.

However, I have some major concerns about the methodology and the results, which I think need to be addressed before the paper can be published.

1. The application of the BNN model is not well described and does not seem to have been very well implemented. The key reference (or equations) is missing for the model. How the model treats dependent (e.g. temperature and precipitation) or independent predictors (e.g. longitude and mammalian fossil type), or maybe conflict predictors are not explained. There is no mention of the issue of collinearity in the selection of the predictors, and whether this was assessed. Some important predictors are not included (see my next point). For a general reader, a brief discussion on the advantage of BNN compared to other ML methods such as random forest (Hengl et al., 2018) will also be interesting.

Response: We thank the reviewer for pointing out these relevant issues. In this revised version, we have now added a more explicit mathematical notation to our BNN model description. We provide this description in the Supplementary Methods, which we reference in the main text in lines 569-571, for readers who are interested in more details about the model. We also provide a fully documented data tutorial to install the software, extract features, train BNN models, and predict vegetation, available on the project's GitHub repository (see "Code availability" section, lines 664-670).

We also added a new paragraph discussing the issue of collinearity and contrasting our approach from mechanistic models (e.g., linear regression), as well as other machine learning models, such as Random Forest (lines 308-327). In short, given the large number of parameters in Neural Network models, collinearity of predictors is generally not of concern, particularly because we are not interested in estimating individual parameter in our model (as is e.g. the case in mechanistic models).

2. I think the greatest problem the current experimental protocol suffers is the lack of some important predictors. For instance, the elevation/landscape, the seasonal features in climate variables, and, considering the deep time, the atmospheric composition.

The uplift of cascades can cause higher elevation cooling that preserves snowpack, and also orographic precipitation and perennial rivers. These factors (by preserving forests) can further affect grassland expansion near the mountain slopes in the midwestern North America (see an overview in Kukla et al., 2021), and in these regions are the boundaries between the open and closed habitats and have the highest uncertainties. A parameter describing elevation/landscape should be taken into account.

I would also say that annual mean temperature and precipitation might not be the crucial predictor for vegetation, and it would be more reasonable to look at the real controls on vegetation, e.g., coldest month temperature, growing season warmth and plant-available moisture. So, my suggestion would be to include summer and winter temperature and precipitation as predictors.

Also, given the time horizons explored in this study, that vegetation can adapt to different atmospheric compositions, particularly the stomata density and photosynthesis/respiration acclimation to varying CO₂ levels, which can potentially modulate the vegetation type. So pCO₂ can also be included as a predictor, or be tested.

Response: We agree that there are many additional suitable predictors that would be very useful for this task. However, the limitation is usually that these predictors are not available throughout the whole timeframe covered in this study (last 30 Myr) and/or are not spatially explicit (i.e. available as a spatial grid). This is e.g. the case for the suggested seasonal features in climate variables and for most elements of the atmospheric composition. After an extensive search of additional suitable predictors, we decided to add two additional predictors to our models, for which we found suitable data spanning the whole timeframe of this study: elevation and atmospheric CO₂ content. We agree with the reviewer that both of these predictors constitute important additions to our model and thank the reviewer for emphasizing this point. We reran all analyses presented in this manuscript, based on this updated set of features. The results are largely consistent with our previous findings but have led to reaching higher prediction accuracies with less complex models than in the previous version (only 1 hidden layer with 8 nodes), which may be attributable to these additional informative predictors.

3. The authors discussed the open habitat transition in the whole North America, and a corresponding 50% threshold suggests a relatively recent expansion of grasslands. However, if the authors use smaller subregions instead, such as the midwestern North America, the timing of the transition (e.g. keep the 50% threshold for this subregion) changes and can lead to unsurprising results. Perhaps while keeping the main conclusion for the whole North America, the authors can add predicted fraction (similar to Fig. 3) also for these subregions and have a more comprehensive comparison with previous studies focusing on smaller regions.

Response: We thank the reviewer for sharing this idea and agree that looking at open vegetation expansion for different areas reveals a more nuanced understanding of the spread of this vegetation type. We therefore downloaded additional shape files delineating North America's ecoregions. The grassland through time plots for all ecoregions can be found on the project's GitHub repository (https://github.com/tandermann/paleovegetation/tree/master/plots/open_veg_through_time/open_vegetation_through_time_per_ecoregion). We added the plot of open vegetation expansion for the American Great Plains ecoregion to the Supplementary Material (Supplementary Figs. S6 and S7). Having this data compiled separately for the American Great Plains allows us in the revised version of the manuscript to discuss the location of early open habitats with more evidence than simply eyeballing the map (see lines 184-188).

4. The uncertainties in the input are completely ignored in the study. I wouldn't regard climate fields from ref24 (is that the right source by the way? I cannot find the 1 deg precipitation and temperature dataset from this paper) as 'reliable' predictors without any tests as they were generated by a model. Linear interpolation of the climate fields from 5 Myr to 1 Myr intervals also seems sloppy. It would be useful to know how much the biases in climate fields can lead to uncertainties in predicted vegetation fraction, say, by perturbing the climate variables systematically (e.g. Lindgren et al., 2021) by $\pm 10\%$, $\pm 20\%$? Maybe the authors have an alternative way of quantifying this, and, if so, it would be useful to include in your results. Additionally, I think the authors should be cautious to use 'reconstruction' (e.g., L259, L275) to describe the probabilistic model results.

Response: The reviewer is correctly pointing out that the predictors used for precipitation and temperature (and also for elevation in the updated manuscript) are based on model predictions and may therefore suffer from biases introduced by these models. To investigate the effect that changes in these predictors may have on the predicted vegetation models, we randomly perturbed the temperature, precipitation, and elevation values of each grid cell (lines 482-495). These perturbations were done on a per-cell basis (not perturbing in a spatially autocorrelated manner), because the predictions made by our models are done on a per-cell basis and are not taking into account the values of the neighboring cells. When predicting vegetation patterns with these modified features in combination with the 105 remaining features, the predicted vegetation patterns are very similar, only showing different vegetation inferences caused by these perturbations for a handful of cells (Supplementary Fig. S9). We also emphasized throughout the revised manuscript the fact that these values are derived from models, e.g., by avoiding referring to these as climate/elevation "reconstructions".

We realized, thanks to the reviewer's comment, that the climate models are not available in the previously cited reference (ref24). While ref24 mentions the data "*Global temperatures were calculated in this manner for 100 Phanerozoic reconstructions of paleo-Köppen belts (one map ~ 5 million years). For an in-depth discussion of the data and methodology used see the Supplementary Materials.*", they are not available in the supplementary material for download. A more appropriate reference for the data is the new ref24 (Scotese, 2021. *An Atlas of Phanerozoic Paleogeographic Maps: The Seas Come In and the Seas Go Out.*). This review reports the data from the latest climate models but does not provide the data for download. The modelled data were provided to us personally by Christopher Scotese, the main author of ref24. In the revised version we are pointing this out explicitly (line 477-481).

Minor points

L65 Please provide key references for the BNN you used.

Response: We added a detailed description in mathematical notation (Supplementary Methods) and provided a link to the GitHub repository containing the associated BNN Python package produced by us (see reply to previous comment).

L85 How does the mean global temperature from ref26 compare with the global mean of annual temperature from ref24?

Response: Both curves show a temperature difference of approximately 6°C between today's global temperature and that of 30 million years ago (see plot below, y-axis shows temperature difference relative to today's temperature). However, the overall trajectory through time differs between these two sources. Part of the discrepancy is likely an issue of temporal

resolution. As the reviewer pointed out correctly in a previous comment, the climate data from ref 24 are derived from models, modeling spatial differences across the globe, but at a coarse temporal resolution (5 Ma steps). The Zachos curve (ref 26) on the other hand is based on data recovered from a few selected deep-sea drilling sites, supplemented by data from the Antarctic, not reflecting the geographic spread of the data summarized from the climate models of ref24, but on the other hand being much more temporally explicit (at least 0.1 Ma resolution). Despite the discrepancy between these two data sources, both predictors constitute relevant input for our models in different ways; while the Zachos curve provides temporally explicit predictor data, the rasters from ref24 provide spatially explicit (modeled) data for our models.

Fig. 1 This figure is useful and contains a lot of information.

Please reorganize it to make it easier to read. Please also add more details, for instance, the colors of taxon nodes, fully connected nodes, and output nodes.

Response: We improved the design of this figure for easier readability. We organized it vertically and delimited the three parts of this figure with differently colored boxes. Further, we added a legend explaining the different types of nodes and their coloration. We also reformulated parts of the figure caption to better explain the figure content.

L113 An alternative for the SYNMAP global potential vegetation data is the pre-industrial vegetation reconstruction, e.g., LandCover 6k biome reconstructions (Harrison et al., 2020, Figure 7)

Response: We thank the reviewer for pointing out this interesting resource. Particularly the data on open vegetation cover at 6,000 and 200 years ago could also be useful for future implementations of similar models, trained on much shallower time-frames than the one covered in our study. For the purposes of our study, we are using SYNMAP, since it provides complete spatial coverage (spatial grid) for all of North America, making it a very comprehensive data source to subsample current vegetation points and evaluate the estimated current vegetation pattern.

L155 Is the 20% randomly selected?

Response: Yes, in each CV iteration a different 20% of the instances are randomly selected as test set. In general, instances are always shuffled before selecting training and test set, to

ensure breaking any potential autocorrelations between neighboring instances, with sometimes similar feature values.

Fig. 3 Maybe 5 Myr interval should be highlighted, so the readers are aware of the interpolation of climate fields.

Response: The climate and elevation fields used in our models only represent a small fraction of the total number of features (3 of 108 features). Therefore, their coarse spatial resolution of 5 Myr intervals does not affect the overall temporal resolution of the model predictions, which are mostly based on the other features with a finer spatial resolution. We therefore would prefer not to specifically emphasize these intervals in the figure, since it may confuse readers to think the predictions are in coarser time units than they actually are. However, we spaced the x-axis tick-labels in 5 Myr units, which helps in assessing the predictions in regard to the temporal resolution of the climate and elevation fields.

L230 & Table 1 I am confused here. Is this outperformance reflected by the table?

Response: We reformulated this section to make it less confusing which comparisons are being made and updated it with the new results. Now this section reads as follows: “*We find that a mixed-feature model that is trained on both biotic and abiotic features (model 1) outperforms a model that is trained on abiotic features only (model 4) by a margin of 3.6% prediction accuracy (Table 1).*”. This result is reflected in Table 1 (see Accuracy column and compare model 1 with model 4).

Fig. 4 how to interpret the values of importance? What does it mean when it's negative? The importance of the 3rd to 15th features seems very comparable.

Response: Feature importance values reported here represent the impact that a given feature has on the prediction accuracy of a model. A negative value means that the model's prediction accuracy actually increases when the feature is dropped (randomized). In the revised manuscript we plot the mean values in Fig. 4, instead of the whole confidence interval. This helps to focus on the mean posterior estimate of feature importance of each feature, showing that a number of features has a delta-accuracy of close to 0, but none are negative. There is also more variation between the top 15 features in the updated results. For increased clarity we added more explanation about the interpretation of the feature importance values to the figure caption (lines 280-285).

L325 Each stage has a long span, and I cannot find how the mean age is calculated from max and min ages (e.g. in supplementary data S2).

Response: In the earlier draft the “mean_age” column actually contained a date drawn randomly between the min and max age of each occurrence (we missed updating the column header after deciding to take this approach). However, in the revised manuscript we decided to use the true mean-ages instead (see “Age Mean” in Supp. Data S2), rounded to the midpoint of the encompassing geological stage (“Binned Age”).

L328 Some of the vegetation types are underrepresented in the data, e.g. the desert. If there is desert (not in Table S1 though), did you classify it as open habitat?

Response: Yes, we classified desert vegetation as open habitat. We provide some more detail about what we considered as “open” vs. “closed” in lines 87-88 and in the updated Table S1 (see where “Vegetation Interpretation” equals “Desert vegetation”).

L391 Just like the vegetation cover is affected by human land alterations (L114), the occurrences of biodiversity are also changed by human activities. But there are no ‘potential’ biodiversity maps.

Response: This is a good point. The assumption that we’re making here is that the current distributions of species are still indicative of their natural habitat associations. We consider this assumption as justified, unless there is a substantial number of species who change their habitat preference due to human impact (e.g., former open grassland species become associated with forest habitats to evade human pressure). While this may be happening on small local scales and while many geographic ranges of species have been severely altered by human impact, we still believe there is reason to assume that current habitat associations of species are indicative of their natural habitat preference. We stated this assumption explicitly in the revised manuscript (lines 469-475).

L410 Why not also interpolate climatic predictors to midpoint of each geological stage?

Response: The interpolation to the midpoint of geological stages is a methodological necessity for the occurrence data, since these need to be temporally binned in some fashion. On the other hand, all abiotic data are used at higher temporal resolution in our models. It is methodologically most convenient to have these data present at a resolution that at least matches the temporal spacing of the predicted vegetation maps (1 Myr), which is why we decided to interpolate the climate and elevation fields at 1 Myr intervals.

Reviewer #2 (Remarks to the Author):

This is an exciting study that addresses a classic scientific question – when and where did open vegetation in North America began to emerge over the last 30 million years – through the combination of deep learning models and a compilation of paleoecological data from across an array of paleobiological and biological data resources. The paper is a technical tour de force, presenting in detail a Bayesian Neural Network (BNN) to infer past vegetation distributions from mammal data. The assembly of paleovegetation and mammalian fossil occurrence data from across a range of paleodata resources is also impressive. The compilation of paleovegetation data is particularly valuable.

The description of the BNN structure and results are clear and thorough. Note that I am not a Bayesian statistician myself, but I have collaborated with them. So I cannot give a deep assessment of the modeling approach, but I read the model description closely and everything made sense to me.

In some ways, the key question posed by this paper (did open vegetation first form 28-23 Ma vs. 23-16 Ma?) isn’t definitively answered, but this paper now clearly establishes where we can confidently see the establishment of open vegetation (after 23 Ma) and where there is persistent uncertainty (before 23 Ma). And the maps of open and forested vegetation over the last 30 Ma (Fig. 2) are fantastic. The paper’s major advances are 1) bringing the most data to bear yet on this question and 2) its innovative methodological approach towards integrate many different lines of paleoecological evidence into one unified inferential synthesis.

Response: We thank the reviewer for the positive and thorough assessment of our manuscript. It is true that based on our results we cannot exclude the presence of open habitats prior to 23 Ma, a point which we now discuss more explicitly in the revised manuscript (see paragraph starting at line 329).

I have a couple of moderate comments about the data analysis and visualizations. The code needs to be made openly available and the data need better documentation – see below for more information.

Response: All scripts and data (except for the paleo-climate and elevation models) used in this study are now available on the project's GitHub repository at <https://github.com/tandermann/paleovegetation>. Additionally, the BNN code is available as a stand-alone Python package (<https://github.com/dsilvestro/npBNN>), to encourage the application of BNNs in future studies. We added this information to the “Code availability” section in the revised manuscript (lines 664-670).

1. Using Geographic distance to calculate spatiotemporal distance is not great (L421-423), especially for a dataset that spans so many latitudes, where the km per degree longitude varies by a factor of ~50%. It would be better to reproject all spatial coordinates onto e.g. Albers equal area grid and then do the distance calculations. It's a pretty important issue, because these spatiotemporal distances are the fundamental input used by the BNN. I would personally recommend recalculating distances and rerunning of the model. However, I recognize this would be a lot of work, so the other non-preferred but possible option would be to simply note in the text that this is a known issue.

Response: We thank the reviewer for making this important point. Accordingly, we converted the coordinates of all taxon occurrences as well as the coordinates of all vegetation points into the Albers equal area projection (EPSG code 5070, sensu <https://guides.library.duke.edu/r-geospatial/CRS>), before using these coordinates to calculate spatial distances (lines 510-512). We reran all analyses that are presented in the revised version using these new distances.

2. L433-436 (& Fig 1): The use of the BNN to come up with a combined weighting of spatial and temporal distance is very interesting. I'd be quite interested to see a table or other summary of those weights, to get a sense of how much relative weight the model gives to spatial distance vs. temporal distance.

a. L486-488: Now I see that all values are collapsed to a single spatial weight and single temporal weight. I'd be curious to see those.

Response: Given the interconnected nature of nodes in the neural network, individual weights don't have a discernible meaning and therefore can't be interpreted by themselves. Below are for example two density plots showing the distribution of MCMC samples for the spatial weight (blue) and the temporal weight (green). The x-axis shows the actual values of those weights, and while the weight for the space distances of all taxa is negative, the weight of the temporal distances is positive and closer to 0. However, these values have no deeper biological meaning and are optimized in context of the weights in the preceding and the following layers. One interesting way of data visualization is to plot the feature maps resulting from this weighing of spatial and temporal distances, showing the map of spatiotemporal vicinity of each taxon (see second figure below, showing the example of the genus *Mammuth*, including the American mastodon). These maps don't show the temporal or spatial weighing, but instead their combined effect in combination with the taxon-specific weights. We added some examples of these maps for visualization purposes to the overview in Fig. 1.

Spatiotemporal “proximity” of Mastodonts for each point on the current map of North America

3. Given that this paper is primarily presenting a new modeling approach, the code needs to be made open and available, both for reproducibility purposes and to help advance the field. Deposition in an open GitHub repository would be sufficient, with a branch version representing the state of the code at the time of the ms.

Response: All code is now made available on GitHub (see our reply to previous comment above).

4. More documentation of the data is needed, so that the reader can better judge the strength of the data underlying these findings, and for reproducibility.

a. Add maps in SuppInfo that show the distribution of a) paleovegetation data points and b) mammal/plant occurrence data points for each time period.

Response: A very good idea, we agree that this will be useful for readers to assess the data coverage through time and space. We added an additional supplementary figure (Fig. S2), showing one map per geological epoch with all taxon occurrences and paleovegetation data points. High resolution versions of these maps can be found on the project’s GitHub repository:

(https://github.com/tandermann/paleovegetation/tree/master/plots/training_data_through_geo_stages).

b. Possibly could add a second set of maps that code the paleovegetation data points by dataset type (pollen/spores, macrofossils, etc.)

Response: We considered this option but given that the dataset type information is not coded categorically, but rather descriptive in some case (see “Fossil type” columns in Supplementary Data S1), and given that even if coded categorically it would result in at least 5-10 categories (depending on the detail), we decided not to plot the data in this manner for now. Note, that our models do not distinguish between dataset type, but treat all paleovegetation information equally. However, in case the reviewer prefers to have these maps added, we are happy to recode the dataset type info into broader categories and plot them on the maps.

c. Upon acceptance of ms., Tables S1 and S2 should be moved out of their current Google Drive and a current permanent archive such as Dryad.

Response: Both supplementary data files (S1 and S2) are available on the permanent Zenodo repository <https://doi.org/10.5281/zenodo.6492100>.

d. Supplementary Figure S1: Color-code the histogram so that for each time period, the proportion of open and forested vegetation is shown.

Response: We added a separate histogram for open paleovegetation points and closed paleovegetation points, respectively (Fig. S1).

5. Supplementary Table S1: This is impressively detailed! A great resource. However, many columns in the table are cryptic or incomplete, so the table needs an explanatory legend or readme file. Specific comments for each column:

a. A/B: Lat/Lon: Specify units (DD)

Response: Units are decimal degrees. This now specified in the column header.

b. C/D: Age Max/Min: Specify units: Ma?

Response: Units are Ma (Mega annum). This now specified in the column header.

c. E/F: Author name/Title/Year: OK but a DOI would be even better. Citation indexers would have a hard time scanning this table and retrieving citation usage.

Response: We have added a column with the DOI/alternative number whenever we could find such a number.

d. 'Entered by': Author codes (T, K, ...) are cryptic. Either show names in full or provide a lookup table somewhere.

Response: This column was for internal purposes only and is not necessary to replicate the study. It has therefore been deleted.

e. 'Reference #:' what does this store? Seems to hold book titles. What does an empty field indicate?

Response: This column duplicates information elsewhere. It has therefore been deleted.

f. 'Locality name': Is this a spatial place name or a geostratigraphic formation name?

Response: Place name. This has been clarified in a footnote, signified by "***". Also, the three columns that contain locality information, such as alternative sample numbers have been labeled as "Locality Name or Number in Publication" I, II, and III.

Footnote: ** These columns contains locality names, numbers and alternative assignments to designate a sample/flora/assemblage in the publication.

g. Depth/Location: Units? Some fields have units and some just have unitless numbers.

Response: We have added units to this column and removed some erroneous numbers.

h. For Columns N onwards, which of these represent information drawn straight from the original publication and when is it based on interpretation of the synthesis authors?

Response: Yes, the information are from the publications themselves. Also, we have reduced the number of columns to only include those that are relevant. We added the following statement as a footnote to the table: "All information in this table is derived from the referenced publication (see column "Publication Name"), unless specified otherwise."

i. R: Title ('Colors match') doesn't match field content, which is country names.

Response: This was an error; it has been changed.

j. What's the difference between Columns AR & AS, which both indicate C3 vs C4?

Response: No real difference, and we have removed the columns as they are not relevant for the paper.

Openness column: We have added an asterisk (*) here, with the footnote: * Openness determination is based on the determination of the author(s), but typically a dominance of fossils of (open-habitat) grasses or other openness indicators (e.g., chenopods, Asteraceae for pollen; small, dry-adapted leaves for megafossils). See discussion in Jacobs et al. (1999) and Strömberg et al. (2018).

6. Supplementary Table S2: Needs more information.

a. Where do all these fossil taxa come from? They are unprovenanced. Need database name, and original publication. Original dataset unique ID would also be helpful.

Response: All fossils were downloaded from public databases. We added “Database Name” and “Publication Name” as additional columns to the table, the latter containing the reference to the original publication.

b. Add a column indicating whether a given taxon name is a mammal or plant (most readers will know, but will help clarify)

Response: We added an additional column “Kingdom” that specifies for each records whether it belongs to Animalia or Plantae.

c. Columns B:G need units

Response: We added units to all columns containing numeric values.

LINE-BY-LINE COMMENTS

L8: already->by

Response: Done.

L28: Here and elsewhere: the reference to grasslands as the most extensive terrestrial biome ‘today’ overlooks the widespread anthropogenic conversion to agriculture. Need to either note agricultural conversion or shift ‘today’ back at least a few centuries.

Response: In the revised manuscript, we mention explicitly that we are referring to the natural, pre-anthropogenic baseline, when referring to today’s vegetation (e.g., lines 31-32: “*Open grasslands today –referring to their natural, pre-anthropogenic extent– represent the most extensive terrestrial biome on Earth,..*”).

L35: Unclear antecedent for ‘their’

Response: Fixed.

L64-69: Clarify in this opening description that you are training the BNN against independent vegetation datasets, both paleo and modern. Took me a deep read before I understood this basic structure of the modeling approach.

Response: We thank the reviewer for this comment, and realize that this was not made sufficiently clear in the initial description. In the revised manuscript we added the following sentence (lines 70-71): “*We train the model using independent vegetation datasets, including paleovegetation and modern vegetation information.*”

L69-75: Also add to this opening paragraph 1-2 sentences that state that you train this model against newly built paleovegetation (N=331) and fossil mammal/plant occurrences (N=7500). This data compilation is as impressive as the model itself, and is a major contribution of this paper.

Response: We agree that this is noteworthy and added two sentences describing the data that was compiled for this study (lines 71-75).

L196 hyphenate ‘here-inferred’

Response: Done.

L278-291: The discussion here could use sharpening: Should this paper’s findings be interpreted as ‘no open vegetation during the Late Eocene (i.e. nothing before 23 Ma, Fig. 3)’ or ‘not enough data to assess whether there were open habitats during the Late Eocene?’

Response: We added a more explicit discussion about our empirical results (lines 329-343), particularly focusing on our findings in regard to the emergence of open vegetation: “*While our model predictions don’t exclude the presence of open vegetation prior to the Miocene,*

the presence of open habitat can only be inferred with confidence starting at the very beginning of the Miocene (23 Ma), based on the 95% HPD interval generated by our best model.”

L292-299: Might note here that the effect of temperature on open vegetation did not emerge until the Pleistocene (1Ma), when Pliocene-Pleistocene cooling resulted in a large area of open tundra across the Arctic (Fig. 1Ma). So the temperature effect is there but later/secondary.

Response: In the revised version of the manuscript we discuss this connection more explicitly in lines 339-341: “Further, our model predictions place the peak expansion rate of open vegetation at the Pliocene-Pleistocene transition (2-3 Ma). This time-period is characterized by a global drop in temperatures and the onset of glacial-interglacial cyclicality³⁸, putatively explaining the disappearance of forests in the northern part of the continent, and leading to an expansion of open vegetation in these areas (Fig. 2).”

L340: Was this random choice of open vs. closed change for these three sites allowed to vary among analytical runs, or was it chosen once and then kept fixed for all analyses?

Response: Upon checking the raw data we realized that none of these ambiguous vegetation points had made it into the final dataframe that was used in this study (after filtering out spatiotemporal duplicates as described in the methods). Therefore we removed this sentence altogether from the manuscript, as it doesn't apply to the data presented in this study.

Abstract
Some of the most extensive terrestrial biomes today consist of open vegetation, including
temperate grasslands and tropical savannas. These biomes, in their current form, originated
relatively recently in Earth's history, likely replacing forested habitats in the second half of the
Cenozoic. However, the precise timing of their origination and the dynamics of their expansion in
different continents remain disputed. Here, we present a probabilistic model of high-resolution
paleovegetation changes in North America, showing that open habitats originated at around 23
million years ago (Ma) in the central part of the continent and increased through time, with an
accelerated expansion rate starting at 5 Ma. By the time of the onset of the Pleistocene glacial
cycles, open habitats were already covering more than 30% of the North American continent and
were expanding at peak rates, to eventually become the most prominent natural vegetation type
today. Our new Bayesian deep learning model integrates comprehensive information from fossil
evidence, geologic models, and paleoclimatic proxies. The model is trained to predict vegetation
type based on associations between plant communities and multiple biotic and abiotic predictors,
including proximity to mammalian and plant macrofossil occurrences, latitude, estimates of
temperature, precipitation, and elevation, as well as the effects of spatial and temporal

[revised manuscript text omitted]

148

A Extracting biotic & abiotic features

Vegetation	Lon	Lat	Age
?	-141	67	2.84
	-146	64	4.47
	-125	60	0

▶ Precipitation = 0.32
 ▶ Temperature = 0.41
 ▶ Elevation = 0.28
 ▶ Temp (global) at time T = 0.65
 ▶ CO₂ (global at time T) = 0.92
 ▶ Lon = -125, Lat = 60, Age = 0

Abiotic features

	Stage 1	Stage 1	Stage 2	Stage 2	
	Δspace	Δtime	Δspace	Δtime	
Horses (Equus)	1600	0.00	2800	0.07	... ▶
Mastodon (Mammut)	/	/	1300	0.07	17 stages

100 taxa

Biotic distances

B Feature generation

C BNN classifier

Figure 1: The process of feature generation and the BNN model architecture. (A) The workflow is shown exemplarily for one point with current vegetation information (framed in red), located at the coordinates -125, 60 (Decimal Degree System) and labelled as “closed” vegetation. Our database, compiled for this study, contains other points with current or past vegetation information, labelled as open (grass symbol) or closed (tree symbol). Once the model is trained it can be applied to estimate the vegetation interpretation for points in space and time, which are currently lacking such information (represented by the question mark). For the selected point, defined by its longitude (Lon), latitude (Lat), and age, we extract several abiotic features, reflecting climatic, geographic, and temporal variables (see box “Abiotic features”). Additionally, we extract the spatial distance to the closest occurrence of each taxon in our occurrence database (see box “Biotic distances”). This is repeated for each geological stage (n=17), while also extracting the temporal distance between the given point and the mid age of each geological stage. (B) These spatial and temporal distances, extracted separately for 100 mammal and plant taxa, are the input of the first two hidden layers in the BNN model. During training, the BNN optimizes weights (represented by lines labelled with w_{XX}) to reduce the multitude of spatial and temporal distance measurements into one single “proximity” value for each taxon (taxon nodes) relative to the given point in space and time. This process of feature generation is equivalent to the convolutional layers in an image classifier, reducing higher-dimensionality data into lower-dimensionality features for input into the subsequent neural network layers. (C) The taxon node values (“Biotic features”) are then used in combination with the abiotic features as input into the fully connected BNN classifier layers. Jointly with the weights of the feature generation layers, the weights of the BNN classifier are estimated during training through MCMC sampling, to optimally map the input data to the correct output vegetation label (“open” or “closed”). Once trained, a posterior sample of the weights is stored for each model and is used to make vegetation predictions for points with unknown vegetation interpretation.

Table 1. Prediction accuracy of tested model configurations. The overall accuracy of each model constitutes the weighted mean between the paleovegetation accuracy (factor 10) and the current vegetation accuracy (factor 1). This weighing was applied because the paleovegetation data are distributed across 10 geological stages, while the current accuracy only represents 1 single stage. 
[revised manuscript text omitted]

Acknowledgements
We thank all paleontologists who have produced and made publicly available the paleovegetation
data and fossil occurrence data necessary for training the models presented in this study,
Christopher Scotese for providing climate models for paleotemperature and paleoprecipitation,
Thomas Alan Neubauer for advice on the paleoclimate and elevation models, Juan Carrillo for
providing paleontological feedback on the mammal fossil data, and two anonymous reviewers and
the editor Devin Ward for helpful feedback on earlier drafts of this manuscript. TA and DS
received funding from the Swedish Research Council (2019-04739). TA was supported by the
SciLifeLab & Wallenberg Data Driven Life Science Program (grant: KAW 2020.0239). DS
received funding from the Swiss National Science Foundation (PCEFP3_187012). AA
acknowledges financial support from the Swedish Research Council (2019-05191), the Swedish
Foundation for Strategic Research (FFL15-0196), and the Royal Botanic Gardens, Kew. CAES
acknowledges the United States National Science Foundation (EAR-1253713). All computations
were carried out on the Kebnekaise computing cluster, as part of the High Performance Computing
Center North (HPC2N), which is funded by the Swedish National Infrastructure for Computing
(SNIC), as well as the Kempe Foundations and the Knut and Alice Wallenberg Foundation.
Data Availability
The supplementary material accompanying this manuscript contains Supplementary Text,
Supplementary Figures S1-S9, and Supplementary Table S1. Additionally, Supplementary Data
(S1 and S2) are available in the Zenodo repository <https://doi.org/10.5281/zenodo.6492100>. The
repository also contains all datasets analyzed and generated during the current study, as well as
source data for all figures and tables. The paleotemperature and paleovegetation data through
time used in this study are published in Scotese, 2021²⁴, but restrictions apply to the availability
of these data, which were used under license for the current study, and so are not publicly
available. Data are however available from the authors upon reasonable request and with
permission of Christopher Scotese.
Code Availability
All code used in this study, as well as a full data tutorial and installation instructions for training
BNN models and predicting vegetation through time, are available on the project's GitHub
repository (<https://github.com/tandermann/paleovegetation>, v1.0.0). The main BNN
functionalities can be loaded as a stand-alone and open-source Python package, which is available
on GitHub (<https://github.com/dsilvestro/npBNN>, v0.1.12), allowing the application of the
described BNN approach for any classification or regression task, not only restricted to the task of
vegetation prediction.
Author Contributions
TA, AA, and DS contributed to conception and design of the study. CS compiled and revised the
paleovegetation data. TA compiled all other data, wrote the code with contributions from DS, ran
all analyses, and wrote the first draft of the manuscript with contributions from all authors. All
authors contributed to the article and approved the submitted version.

Reviewers' Comments:

Reviewer #1:

Remarks to the Author:

In the revised manuscript, I appreciate the thorough discussion and justification of the BNN modelling approach, and the careful testing of additional predictors, and their uncertainties. I believe that the authors have responded to my comments in a satisfactory fashion, and see no further obstacle to publication of the manuscript, pending a few technical issues.

1. In the reply, the authors indicated some potential important predictors but limited by timeframe or spatial coverage. This message can be integrated to Method and the readers are aware of this potential weakness, and it may stimulate future studies on expanding the existing datasets.
2. As the authors pointed out, the modelled temperature and precipitation dataset was provided by personal correspondence, so I am unable to fully assess.
3. The results of uncertainty test are interesting (new Fig. S9) and highlight the robustness of the approach. The model sensitivity to randomly perturbed predictors is not obvious for the current vegetation map. I am wondering if this is a similar situation for past time periods.

Reviewer #2:

Remarks to the Author:

The revised manuscript has done a very good job of addressing my comments on the first version of the manuscript. Thanks to the authors for their careful work. I have no further major comments. I look forward to seeing this paper in print.

A few last minor requests:

*On Lines 71-73, where the dataset compilation is now introduced, add pointers to Supplementary Data, Paleobiology Database, Cenozoic Angiosperm Database, GBIF, and PALEOMAP so that reader has a good initial sense of some of the major data sources underpinning this analysis. The full workup in the Methods section is good.

*I have made a few minor edits in the PDF version of the manuscript that was added to the end of the authors' response.

*The journal and book titles in the references are inconsistently capitalized.

Reviewer #1 (Remarks to the Author):

In the revised manuscript, I appreciate the thorough discussion and justification of the BNN modelling approach, and the careful testing of additional predictors, and their uncertainties. I believe that the authors have responded to my comments in a satisfactory fashion, and see no further obstacle to publication of the manuscript, pending a few technical issues.

1. In the reply, the authors indicated some potential important predictors but limited by timeframe or spatial coverage. This message can be integrated to Method and the readers are aware of this potential weakness, and it may stimulate future studies on expanding the existing datasets.

Response: We added this information into the Methods section (lines 498-505).

2. As the authors pointed out, the modelled temperature and precipitation dataset was provided by personal correspondence, so I am unable to fully assess.

Response: We regret not being able to make this fully available, since these specific data don't belong to us or the public domain. But we are confident that interested readers will be able to receive the data upon request from Chris Scotese, just as we did.

3. The results of uncertainty test are interesting (new Fig. S9) and highlight the robustness of the approach. The model sensitivity to randomly perturbed predictors is not obvious for the current vegetation map. I am wondering if this is a similar situation for past time periods.

Response: Yes, the predictions also for past periods remain largely unaffected by the perturbations. We added Supplementary Fig. S10 showing the comparison between past predicted vegetation maps with the original feature set compared with those produced with the perturbed feature set. We also paste a small version of that figure below.

Reviewer #2 (Remarks to the Author):

The revised manuscript has done a very good job of addressing my comments on the first version of the manuscript. Thanks to the authors for their careful work. I have no further major comments. I look forward to seeing this paper in print.

A few last minor requests:

*On Lines 71-73, where the dataset compilation is now introduced, add pointers to Supplementary Data, Paleobiology Database, Cenozoic Angiosperm Database, GBIF, and PALEOMAP so that reader has a good initial sense of some of the major data sources underpinning this analysis. The full workup in the Methods section is good.

Response: We added these references to the paragraph in the introduction. We agree that this provides readers a good initial overview.

*I have made a few minor edits in the PDF version of the manuscript that was added to the end of the authors' response.

Response: Thank you for those edits, we implemented them all in the main manuscript file.

*The journal and book titles in the references are inconsistently capitalized.

Response: We carefully went through the reference list and fixed all inconsistencies.